# Learning representations for image-based profiling of perturbations

Nikita Moshkov[1], Michael Bornholdt[2], Santiago Benoit[2,3], Matthew Smith[2,4], Claire McQuin[2], Allen Goodman[2], Rebecca A. Senft[2], Yu Han[2], Mehrtash Babadi[2], Peter Horvath [1], Beth A. Cimini [2], Anne E. Carpenter [2], Shantanu Singh [2] & Juan C. Caicedo[2,5,6] ✉

Measuring the phenotypic effect of treatments on cells through imaging assays is an efficient and powerful way of studying cell biology, and requires computational methods for transforming images into quantitative data. Here, we present an improved strategy for learning representations of treatment effects from high-throughput imaging, following a causal interpretation. We use weakly supervised learning for modeling associations between images and treatments, and show that it encodes both confounding factors and phenotypic features in the learned representation. To facilitate their separation, we constructed a large training dataset with images from five different studies to maximize experimental diversity, following insights from our causal analysis. Training a model with this dataset successfully improves downstream performance, and produces a reusable convolutional network for image-based profiling, which we call Cell Painting CNN. We evaluated our strategy on three publicly available Cell Painting datasets, and observed that the Cell Painting CNN improves performance in downstream analysis up to 30% with respect to classical features, while also being more computationally efficient.

High-throughput imaging and automated image analysis are powerful tools for studying the inner workings of cells under experimental interventions. In particular, the Cell Painting assay[1,2] has been adopted both by academic and industrial laboratories to evaluate how perturbations alter overall cell biology. It has been successfully used for studying compound libraries[3–5], predicting phenotypic activity[6–9], and profiling human disease[10,11], among many other applications. To reveal the phenotypic outcome of treatments, image-based profiling transforms microscopy images into rich high-dimensional data using morphological feature extraction[12]. Cell Painting datasets with thousands of experimental interventions provide a unique opportunity to use machine learning for obtaining representations of the phenotypic outcomes of treatments.

Improved feature representations of cellular morphology have the potential to increase the sensitivity and robustness of image-based profiling to support a wide range of discovery applications[13,14]. Feature extraction has been traditionally approached with classical image processing[15,16], which is based on manually engineered features that may not capture all the relevant phenotypic variation. Several studies have used convolutional neural networks (CNNs) pre-trained on natural images[11,17,18], which are optimized to capture variation of macroscopic objects instead of images of cells. To recover causal representations of treatment effects, feature representations need to be sensitive to subtle changes in morphology. Researchers have found that training or fine-tuning networks with high-throughput images can improve downstream performance compared to models trained for

[1]HUN-REN Biological Research Centre, 62 Temesvári krt, Szeged 6726, Hungary. [2]Broad Institute of MIT and Harvard, 415 Main St, Cambridge, MA 02141, USA. [3]Carnegie Mellon University, 5000 Forbes Ave, Pittsburgh, PA 15213, USA. [4]Harvard College, 86 Brattle Street Cambridge, Cambridge, MA 02138, USA. [5]Morgridge Institute for Research, 330 N Orchard St, Madison, WI 53715, USA. [6]Department of Biostatistics and Medical Informatics, University of Wisconsin-Madison, 1300 University Ave, Madison, WI 53706, USA. ✉e-mail: juan.caicedo@wisc.edu

natural images[19–22]. This indicates that representation learning can identify domain-specific features from cellular images in a data-driven way[4,23–26], which also brings unique challenges to prevent confounding factors[27,28].

In this paper, we investigate the problem of learning representations for image-based profiling with Cell Painting. Our goal is to identify an optimal strategy for learning cellular features, and then use it for training models that recover improved representations of the phenotypic outcomes of treatments. We use a causal framework to reason about the challenges of learning representations of cell morphology (e.g., confounding factors), which naturally fits in the context of perturbation experiments[29,30], and serves as a tool to optimize the workflow and yield better performance (Fig. 1). In addition, we adopted a quantitative evaluation of the impact of feature representations in a biological downstream task, to guide the search for improved workflow. The evaluation is based on querying a reference collection of treatments to find biological matches in a perturbation experiment. In each evaluation, cell morphology features change to compare different strategies, while the rest of the image-based profiling workflow remains constant. Performance is measured using metrics for the quality of a ranked list of results for each query (Fig. 1G). With this evaluation framework, we conduct an extensive analysis on three publicly available Cell Painting datasets.

Within the proposed causal framework, we use weakly supervised learning (WSL)[23] to model the associations between images and treatments, and we found that it powerfully captures rich cellular features that simultaneously encode confounding factors and phenotypic outcomes as latent variables. Our analysis indicates that to disentangle them, to improve the ability of models to learn the difference between the two types of variation, and to recover the causal representations of the true outcome of perturbations, it is important to train models with highly diverse data. Therefore, we constructed a training dataset that combines variation from five different experiments to maximize the diversity of treatments and confounding factors. As a result, we successfully trained a reusable single-cell feature extraction model: the Cell Painting CNN (Fig. 1F), which yields better performance in the evaluated benchmarks and displays sufficient generalization ability to profile other datasets effectively.

## Results

### Recovering features of treatment effects

We use a causal model as a conceptual framework to reason and analyze the results of representation learning strategies. Fig. 1B presents the causal graph with four variables: interventions (treatments **T**), observations (images **O**), outcomes (phenotypes **Y**) and confounders (e.g. batches **C**). Some variables are observables (white circles), while others represent latent variables (shaded circles). Note that confounders can include a wide range of technical / nuisance variation, which we group together and refer to as batch effects to be consistent with the related literature. This graph is a model of the causal assumptions we make for representation learning and for interpreting the results.

Our goal is to estimate an unbiased, multidimensional representation of treatment outcomes (**Y**), which can later be used in many downstream analysis tasks. We use WSL[23] (Fig. 1C) for obtaining representations of the phenotypic outcome of treatments by training a classifier to distinguish all treatments from each other. In this way, the model learns associations between observed images (**O**) and treatments (**T**), while capturing unobserved variables in a latent representation (**Y** and **C**). To recover the phenotypic features of treatment effects (**Y**) from the latent representations, we employ batch correction[18] to reduce the variation associated with confounders and amplify causal features of phenotypic outcomes (Fig. 1D). More details of the assumptions and structure of our framework can be found in the Methods section.

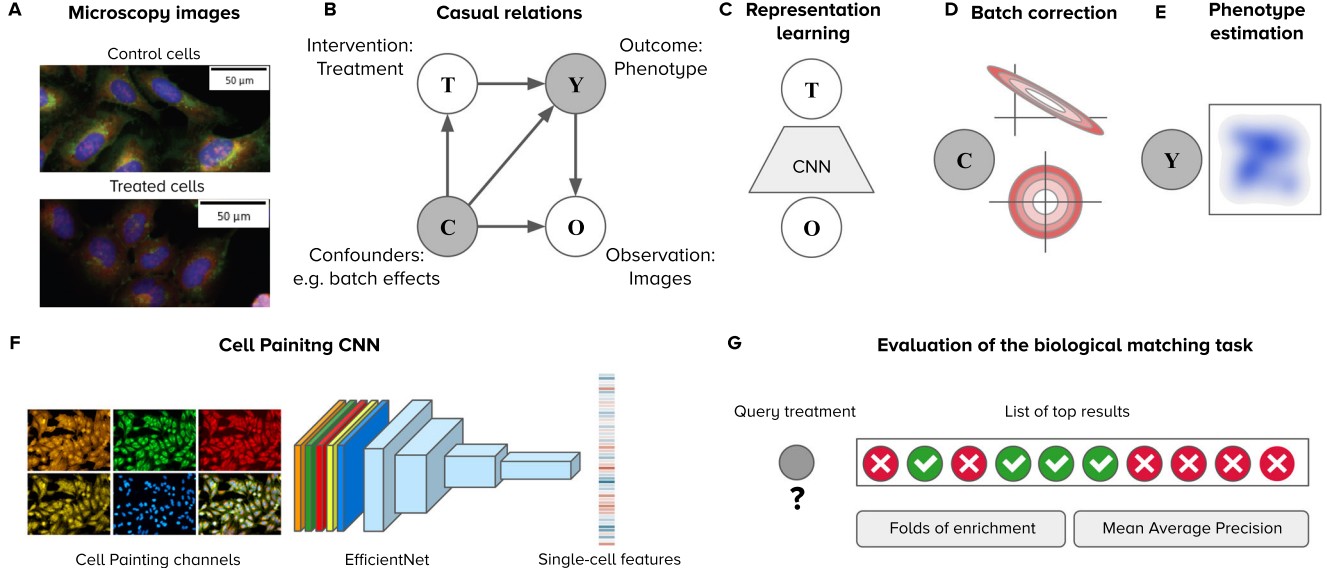

**Fig. 1 | Framework for analyzing image-based profiling experiments. A** Example Cell Painting images from the BBBC037 dataset of control cells (empty status) and one experimental intervention (JUN wild-type overexpression) in the U2OS cell line. **B** Causal graph of a conventional high-throughput Cell Painting experiment with two observables in white circles (treatments and images) and two latent variables in shaded circles (phenotypes and batch effects). The arrows indicate the direction of causation. **C** Weakly supervised learning as a strategy to model associations between images (**O**) and treatments (**T**) using a convolutional neural network (CNN). The CNN captures information about the latent variables **C** and **Y** in the causal graph because both are intermediate nodes in the paths connecting images and treatments. **D** Illustration of the sphering batch-correction method where control samples are a model of unwanted variation (top). After sphering, the biases of unwanted variation in control samples is reduced (bottom). **E** The goal of image-based profiling is to recover the outcome of treatments by estimating a representation of the resulting phenotype, free from unwanted confounding effects. **F** Illustration of the Cell Painting CNN, an EfficientNet model trained to extract features from single cells. **G** The evaluation of performance is based on nearest neighbor queries performed in the space of phenotype representations to match treatments with the same phenotypic outcome. Performance is measured with two metrics: folds of enrichment and mean average precision (Methods).

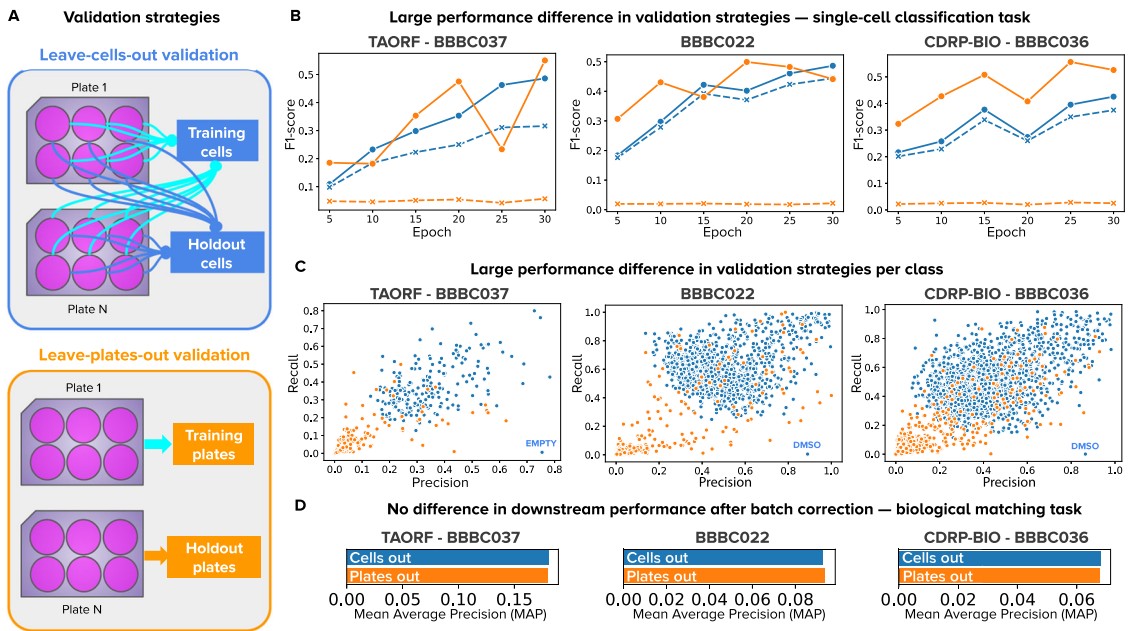

**Fig. 2 | Validation strategies for the single-cell classification task in weakly supervised learning. A** Illustration of the two strategies: leave-cells-out (in blue) uses cells from all plates in the dataset for training and leaves a random fraction out for validation. Leave-plates-out (in orange) uses all the cells from certain plates for training, and leaves entire plates out for validation. Any difference in performance is due to confounding factors. Note that plates-left out are selected such that all treatments have two full replicate-wells out for validation, which may or may not correspond to entire batches, depending on the experimental design. **B** Learning curves of models trained with WSL for 30 epochs with all treatments from each dataset. The x-axis is the number of epochs and the y axis is the average F1-score. The color of lines indicates the validation strategy, and the style of lines indicates training (solid) or validation (dashed) data. **C** Precision and recall results of each treatment in the single-cell classification task. Each point is a treatment (negative controls are labeled in blue), and the color corresponds to the validation strategy. **D** Performance of models in the downstream, biological matching task after batch correction. Source data is provided as a Source Data file.

## Weakly supervised learning captures confounders and phenotypic outcomes of treatments

WSL models are trained with a classification loss to detect the treatment in images of single cells (Fig. 1C, Methods), which is a pretext task to learn representations of the latent variables in the causal graph. Given that the treatment applied to cells in a well is always known, we quantify the success rate of single-cell classifiers on this pretext task using precision and recall. We conducted single-cell classification experiments using two validation schemes to reveal how sensitive WSL is to biological and technical variation (Fig. 2A). The leave-cells-out validation scheme uses single cells from all plates and treatments in the experiment for training, and leaves a random fraction out for validation. By doing so, trained CNNs have the opportunity to observe the whole distribution of phenotypic features (all treatments) as well as the whole distribution of confounding factors (all batches or plates). In contrast, the leave-plates-out validation scheme separates different technical replicates (plates) for training and validation, resulting in a model that still observes the whole distribution of treatments, but only partially sees the distribution of confounding factors.

The results in Fig. 2B show a stark contrast in performance between the two validation strategies. When leaving cells out (results in blue), a CNN can accurately learn to classify single cells in the training and validation sets with only a minor difference in performance. When leaving plates out (results in orange), the CNN learns to classify the training set well but fails to generalize correctly to the validation set. The generalization ability of the two models is further highlighted in the validation results in Fig. 2C, which presents the precision and recall of each treatment.

Importantly, while these WSL models exhibit a major difference in performance in the pretext classification task, their performance in the downstream analysis task is almost the same after batch correction (Fig. 2D). The large difference in performance in the pretext task followed by no difference in the downstream task reveals that WSL models leverage both phenotypes and confounders to solve the pretext task. On one hand, the validation results of leaving-cells-out are an overly optimistic estimate of how well a CNN can recognize treatments in single cells, because the models leverage batch effects to make the correct connection. On the other hand, the results of leaving-plates-out are an overly pessimistic estimate because the CNN fails to generalize to unseen replicates with unknown confounding variation (domain shift). The true estimate of performance in the pretext classification task is likely to be in the middle when accounting for confounding factors.

This is indeed what we observe in the downstream analysis results: after batch correction, the representations of models trained with leave-cells-out and leave-plates-out yield similar downstream performance in the biological matching task, indicating that both models find similar phenotypic features, but capture different confounding variation. Importantly, batch correction is not improving the situation of either of the two strategies compared in Fig. 2D. Instead, batch correction is removing the noise and biases and leveling out the situation of both models. This means that neither of the models learned anything different or more useful than the other despite having drastically different performance on the auxiliary task.

The same effect is observed with alternative WSL approaches. Our experiments are based on an EfficientNet model trained with a classification loss and weak labels. We explored the use of other loss functions that have the potential to improve performance in the presence of weak labels: Online Label Smoothing[31] and Multiple Instance Learning with Attention[32] (Methods). We did not observe improved results when using these alternative WSL formulations (Supplementary Fig. 1), primarily because these methods are designed to improve the performance of a classifier with weak or noisy labels, instead of learning disentangled representations. We use the classifier as a pretext training component, but the main problem to be solved is learning representations that are invariant to confounding factors.

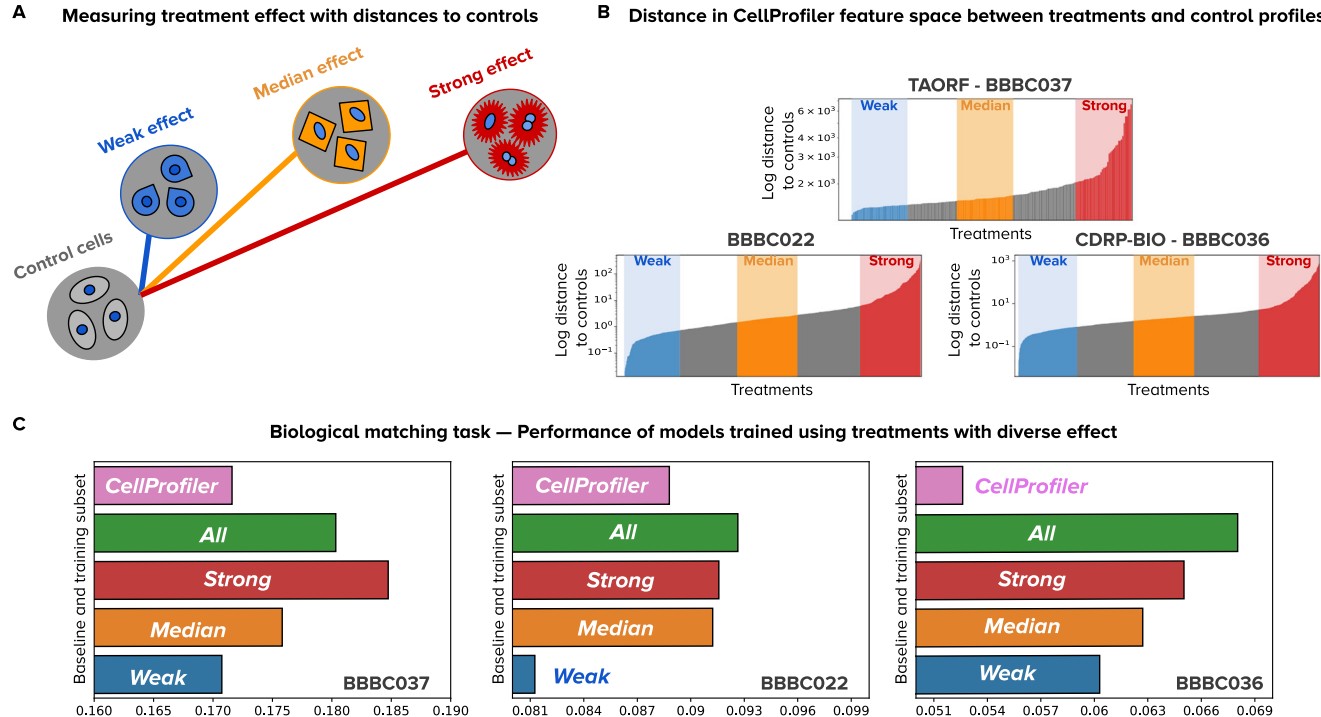

**Fig. 3 | Effect of training models with subsets of treatments. A** Illustration of phenotypic outcomes with varied effects and their distance to controls (see Methods). **B** Distribution of distances between treatments and controls as an estimation of treatment effect sorted by distance for each dataset. The x axis represents individual treatments and the y axis represents the log normalized distance to controls. From this distribution, we select 20% of treatments with the weakest (blue), median (orange), and strongest (red) treatments for experiments. **C** Evaluation of performance in downstream analysis (biological matching task) for each dataset. Each barplot represents one experiment conducted with a model trained with the corresponding subset of the data. The x axis represents performance according to mean average precision (higher is better). Source data is provided as a Source Data file.

Note that using all replicates in the leave-cells-out validation experiment does not result in information leaks with respect to the downstream biological task. Compounds are always known ahead of time, which is the information the models use for training. What we assume to be unknown is the mode of action (MoA), which is information always left out for downstream evaluation and never seen by models during training. This approach measures how well the MoA information emerges from learned representations instead of directly training for MoA classification. Similar evaluation setups have been also used in previous work[23,33].

Technical variation manifests in images in subtle ways that cannot be readily distinguished by eye. There is a set of technical factors that includes microscopes, date and time of acquisition, technician, plate-to-plate variation (differences in assay preparation), well-position effects (differences in humidity and temperature), among others. Despite the best efforts to automate and standardize experiment preparation and image acquisition, these factors continue to influence image-based measurements because they reflect microscopic events that cannot be fully controlled. Some of these factors have stronger effects than others, but all of them accumulate in images in unique ways that represent confounding factors. Technical variation may be recorded in images as hidden patterns that the human eye is invariant to, but computational methods can easily see and incorporate in their metrics. Batch correction methods aim to remove these factors of unwanted variation from image features, and given their unspecified nature, this is still an open research problem.

**Treatments with strong phenotypic effect improve performance**
The WSL model depicted in Fig. 1C captures associations between images (**O**) and treatments (**T**) in the causal graph, while encoding technical (**C**) and phenotypic (**Y**) variation as latent variables because

both are valid paths to find correlations. Given that controlling the distribution of confounding factors does not result in downstream performance changes, in this section we explore the impact of controlling the distribution of phenotypic diversity. Our reasoning is that WSL learning favors correlations between treatments and images through the path in the causal graph that makes it easier to minimize the empirical error in the pretext task. Therefore, the variation of treatments with a weak phenotypic response is overpowered by confounding factors that are stronger relative to the phenotype.

To measure the phenotypic strength of treatments we calculate the Euclidean distance between control and treatment profiles in the CellProfiler feature space after batch correction with a sphering transform (Fig. 3A). We interpret this procedure as a crude approximation of the average treatment effect (ATE), a causal parameter of intervention outcomes, because the Euclidean distance calculates the difference in expected values (aggregated profiles) of the outcome variable (phenotype) between the control and treated conditions. We chose distances in the CellProfiler feature space as an independent prior for estimating treatment strength because these are non-trainable, and because in our experiments CellProfiler features exhibit more robustness to confounding factors (Supplementary Fig. 2).

We ranked treatments in ascending order based on the strength of the phenotypic effect and took 20% in the bottom, middle and top of the distribution (Fig. 3B, Table 1). Next, we evaluated the performance of WSL models trained on each of these three groups and we found that performance improves in the downstream biological matching task with treatments that have a stronger phenotypic effect (Fig. 3C). These results were obtained by training under a leave-cells-out validation scheme, giving the CNN full access to the distribution of confounders. The trend indicates that it is possible to break the limitations of WSL for capturing useful associations between images and

treatments in the latent variables as long as the phenotypic outcome is stronger than confounding factors. Note that training with all treatments (green points in Fig. 3C) may result in lower overall performance if the majority of the treatments have weak phenotypes (BBBC037), or may result in marginally improved performance if the confounding effects are too strong (BBBC022), or may result in better performance when there is a balance between both latent variables (BBBC036). Training with all treatments only improves performance with respect to the CellProfiler baseline in one of the three datasets (Fig. 3D).

Note that strong phenotypic effect (large ATE) is not the same as having high compound concentrations (Methods). Detecting treatments with subtle phenotypic effects is challenging in any platform, including imaging and gene expression[5,34]. Selecting treatments with strong phenotypic effects serves as a strategy to trade-off the sensitivity to subtle biological variation and the introduction of confounding variation. When confounding variation is stronger than subtle phenotypes, WSL models will preferentially rely on the most prominent signal (confounding). Removing treatments with weak phenotypic effect can break this dependency, resulting in models with better performance. The threshold that separates strong vs weak phenotypes was selected in our study based on the percentiles of the ATE distribution, and it can be used as a mechanism to decide how much biological or confounding variation the model observes during training.

## A training set with highly diverse experimental conditions

Training a model with all the data in an individual dataset does not necessarily improve performance with respect to the baseline (Fig. 3C green vs pink points). Changing the distribution of confounders does not impact performance in the biological matching task after batch correction (Fig. 2D). The only factor that impacted performance was changing the distribution of phenotypic outcomes (Fig. 3C). Therefore, we hypothesize that training with data beyond the individual dataset of interest while favoring phenotypic diversity could result in improved performance.

To increase the diversity of experimental conditions in the training set, we created a combined training resource by collecting strong treatments from the five dataset sources listed in Table 2. We first filtered strong treatments from each source and prioritized treatments shared across sources (Methods and Fig. 4). We selected 348 treatments from BBBC022 (strongest 35%), 354 from BBBC036 (strongest 23%) and 47 treatments from BBBC037 (strongest 23%). We complemented these treatments with the corresponding replicates in the LINCS and BBBC043 datasets, and added 7 new compounds and 32 new gene overexpression perturbations, resulting in 488 treatments in total (Fig. 4). This combined set also represents two cell lines, two types of negative controls, and examples from more than 200 plates. This

results in training data with high experimental diversity with respect to the two latent variables in the causal graph: technical variation (confounders **C**) and phenotypic variation (outcomes **Y**).

## Cell Painting CNN learns improved biological features

We found that training a model on this combined Cell Painting dataset consistently improves performance and yields better results than baseline approaches in the task of biologically matching queries against a reference annotated library of treatments, across all three benchmarks (Fig. 5A). We consider two baseline strategies in our evaluation: 1) creating image-based profiles with classical features obtained with custom CellProfiler pipelines (Methods), and 2) computing profiles with a CNN pre-trained on ImageNet, a dataset of natural images in red, green, blue (RGB) colorspace (Methods). Intuitively, we expect feature representations trained on Cell Painting images to perform better at the matching task than the baselines. In the case of CellProfiler, manually engineered features may not capture all the relevant phenotypic variation, and in the case of ImageNet pre-trained networks, they are optimized for macroscopic objects in 3-channel natural images instead of 5-channel fluorescence images of cells.

According to the MAP metric, a WSL model trained on the highly diverse combined set improves performance 7%, 8% and 23% relative to CellProfiler features on BBBC037, BBBC022 and BBBC036 respectively (difference of cyan points vs pink points in the x axis of Fig. 5A). Similar improvements are observed with the Folds of Enrichment metric (y-axis of Fig. 5A), obtaining 6%, 7%, and 30% relative improvement on the three benchmarks respectively. Importantly, the combined dataset allowed us to train a single model once and profile all the three benchmarks without re-training or fine-tuning on each of them, demonstrating that the model captures features of Cell Painting images relevant to distinguish more effectively the variation of the two latent variables of the causal model. We also evaluated the performance of feature extraction with masked cells vs with cells in context, and found that masking cells degrades performance (Supplementary Note 1, Supplementary Fig. 3). All of the experiments reported in our manuscript are based on single-cells cropped in context.

We found that ImageNet features showed variable performance compared to CellProfiler features (Fig. 2B), sometimes yielding similar performance (BBBC022), sometimes lower performance (BBBC037) and sometimes slightly better performance (BBBC036). The three benchmarks used in this study are larger scale and more challenging than datasets used in previous studies[17,18] where it was observed that ImageNet features are typically more powerful than classical features. Our results indicate that, in large scale perturbation experiments with Cell Painting, ImageNet features do not conclusively capture more cellular-specific variation than manually engineered features using classical image processing.

Fig. 5B shows a UMAP projection[35] of the feature space obtained using our Cell Painting CNN for the three datasets evaluated in this study. From a qualitative perspective, the UMAP plot of the BBBC037 dataset (a gene overexpression screen) shows groups of treatments clustered according to their corresponding genetic pathway, and recapitulates previous observations of known biology[36]. The BBBC022[37] and BBBC036[38] datasets (compound screens) likewise show many treatments grouped together

**Table 1 | Subsets of treatments used for model training**

| Dataset | 20% Weak | 20% Median | 20% Strong | All |
|---|---|---|---|---|
| BBBC037 | 41 | 41 | 41 | 205 |
| BBBC022 | 199 | 199 | 199 | 995 |
| BBBC036 | 310 | 310 | 310 | 1,550 |

**Table 2 | Source datasets of the combined training resource**

| Dataset | Plates | Fields of view | Treatment Type | Treatments | Queries | Ground truth | Purpose | Reference |
|---|---|---|---|---|---|---|---|---|
| LINCS | 136 | 23,094 | Compounds | 129 | N/A | N/A | Training only | Way et al.[7] |
| BBBC043 | 16 | 7,992 | Gene ORFs | 50 | N/A | N/A | Training only | Caicedo et al.[10] |
| BBBC036 | 55 | 122,022 | Compounds | 1550 | 1,365 | Mechanism of action | Training and evaluation | Bray et al.[38] |
| BBBC022 | 20 | 66,558 | Compounds | 995 | 849 | Mechanism of action | Training and evaluation | Gustafsdottir et al.[37] |
| BBBC037 | 5 | 17,254 | Gene ORFs | 205 | 204 | Genetic pathway | Training and evaluation | Rohban et al.[36] |

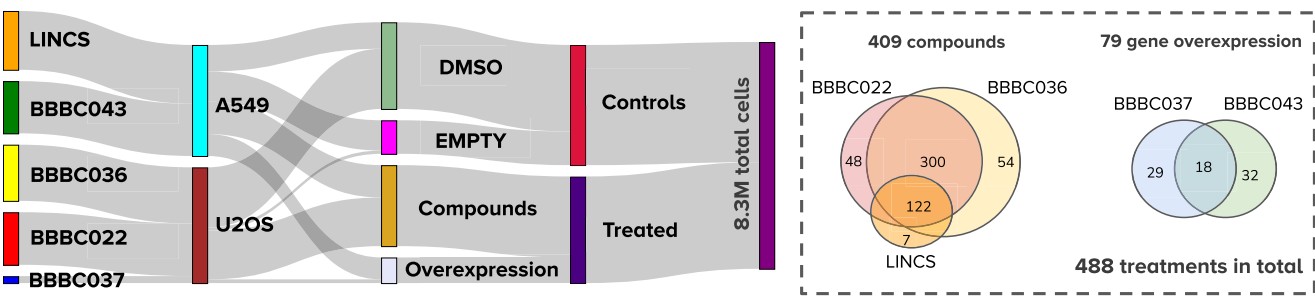

**Fig. 4 | A combined set of Cell Painting images for training.** Statistics of the combined Cell Painting dataset created to train a generalist model, which brings 488 treatments from 5 different publicly available sources (Methods): LINCS, BBBC043, and the three datasets evaluated here; left: Sankey funnel diagram illustrating the distribution of the 8.3 million single cells in this combined dataset. There are two types of treatments (compounds and gene overexpression), two types of controls (empty and DMSO), two cell lines (A549 and U2OS), obtained from 232 plates. Right: the Venn diagrams illustrate the common treatments among dataset sources. Source data is provided as a Source Data file.

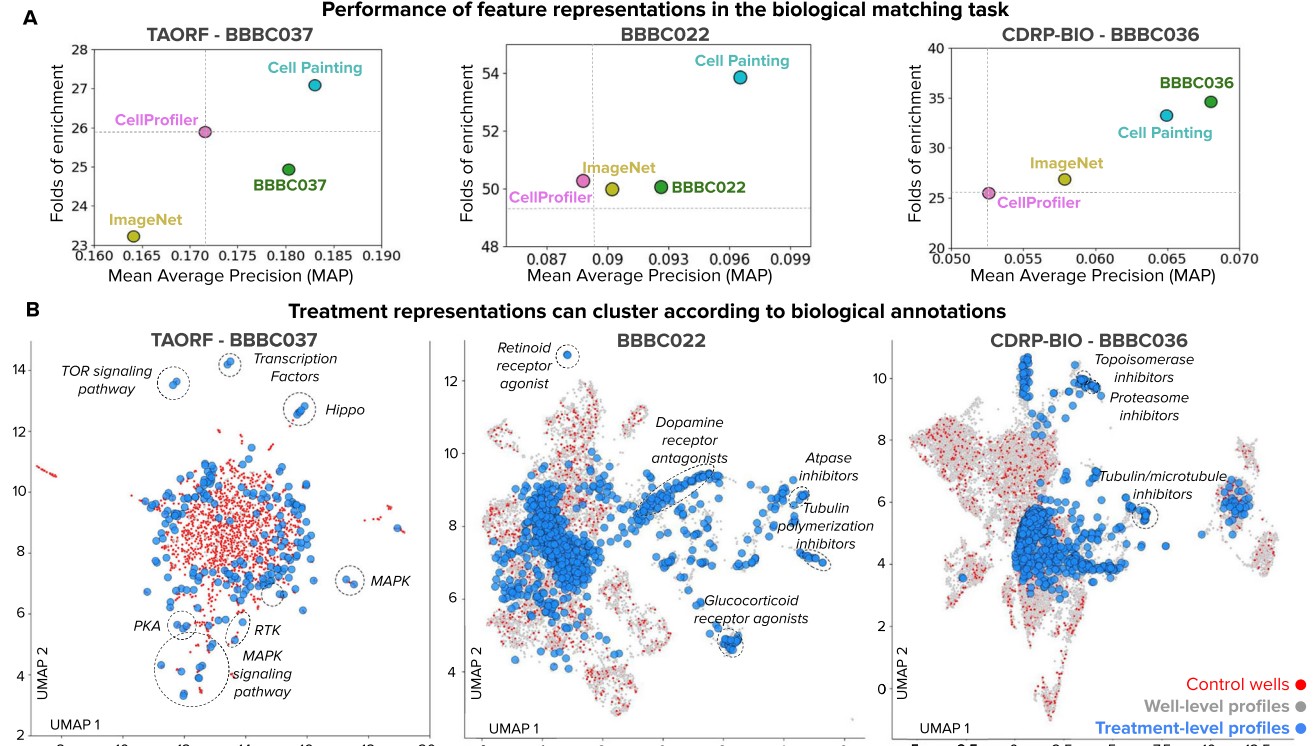

**Fig. 5 | Quantitative and qualitative evaluation of feature representations of treatment effects.** The evaluation task is biological profile matching (see Fig. 1G). **A** Performance of feature representations for the three benchmark datasets according to two metrics: Mean Average Precision (MAP) in the x axis and Folds of Enrichment in the y axis (see Methods). Each point indicates the mean of these metrics over all queries using the following feature representations: CellProfiler (pink), a CNN pre-trained on ImageNet (yellow), a CNN trained on the combined set of Cell Painting images (cyan), and a CNN trained on Cell Painting images from the same dataset (green). In all cases, sphering batch-correction was applied on well-level profiles. **B** 2D UMAP projections of treatment profiles obtained with the Cell Painting CNN (672 features) after batch correction for the three datasets evaluated in this work. The plot includes a projection of well-level profiles (gray points), control wells (red points), and aggregated treatment-level profiles of treatments (blue points). Dashed lines indicate clusters of treatment-level profiles where all or the majority of points share the same biological annotation. Source data is provided as a Source Data file.

according to their mechanism of action (MoA). The clustering is generally consistent across several choices of layers used for feature extraction, resulting in similar groups from a qualitative point of view. Some, but not all, of these clusters are also prominent in the chemical feature space for compound perturbations, suggesting complementarity for correctly connecting MoAs (Supplementary Fig. 4). The quantitative performance metrics offer a more accurate picture of the biological relevance of these clusters (Supplementary Note 3, Supplementary Fig. 8B).

## Batch correction recovers phenotype representations

Batch correction is a crucial step for image-based profiling regardless of the feature space of choice. We hypothesized that a rich representation might encode both confounders and phenotypic features in a way that facilitates separating one type of variation from the other, i.e. disentangles the sources of variation. To test this, we evaluated how representations respond to the sphering transform, a linear transformation for batch correction based on singular value decomposition SVD (Methods). Sphering first finds directions of maximal variance in

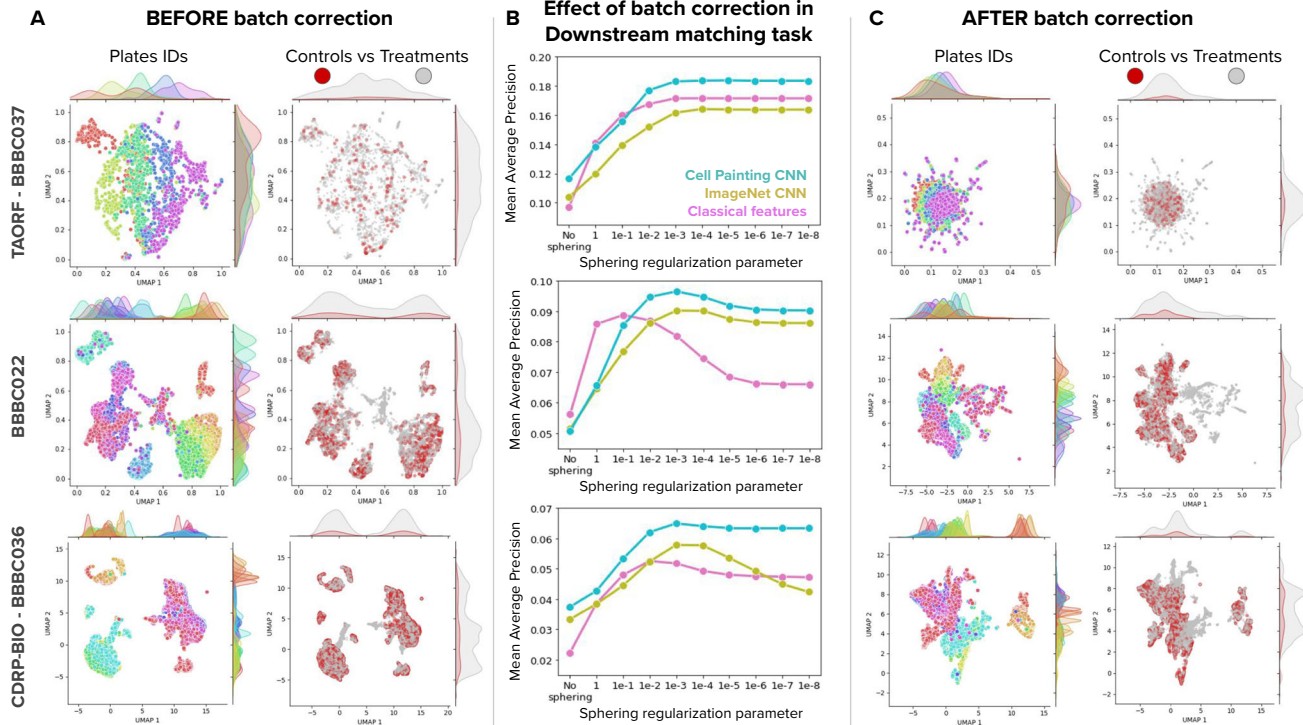

**Fig. 6 | Effect of batch correction on feature representations.** Batch correction is based on the sphering transform and applied at the well-level, before treatment-level profiling (Methods). **A** UMAP plots of well-level profiles before batch correction for the three benchmark datasets (rows) colored by plate IDs (left column) and by control vs treatment status (right column). The UMAP plots display density functions on the x and y axes for each color group to highlight the spread and clustering patterns of data. **B** Effect of batch correction in the biological matching task. The x axis indicates the value of the regularization parameter of the sphering transform (smaller parameter means more regularization), with no correction in the leftmost point and then in decreasing parameter order (increasing sphering effect). The y axis is Mean Average Precision in the biological matching task. **C** UMAP plots of well-level profiles after batch correction for the three benchmark datasets with the same color organization as in (**A**). Source data is provided as a Source Data file.

the set of control samples and then reverses their importance by inverting the eigenvalues. This transform makes the assumption that large variation found in controls is associated with confounders and any variation not observed in controls is associated with phenotypes. Thus, sphering can succeed at recovering the phenotypic effects of treatments if the feature space is effective at separating the sources of variation.

We found that all the methods benefit from batch correction with the sphering transform, indicated by the upward trend of all curves from low performance with no batch correction to improved performance as batch correction increases (Fig. 6B). Downstream performance in the biological matching task improves by about 50% on average when comparing against raw features without correction. The UMAP plots in Fig. 6A show the Cell Painting CNN feature space for well-level profiles before batch correction. When colored by Plate IDs, the data points are fragmented, and the density functions in the two UMAP axes indicate concentration of plate clusters. After sphering, the UMAP plots in Fig. 6C show more integrated data points and better aligned density distributions of plates. The performance of the Cell Painting CNN in the biological matching task also improves upon the baselines (Fig. 6B) and displays a consistent ability to facilitate batch correction in all the three datasets, unlike the ImageNet CNN.

We evaluated two alternative batch correction methods that use nonlinear transformations of features to remove unwanted variation: Harmony[39] and Gradient Reversal Layer[40] (GRL). The results indicate that these nonlinear batch correction methods do not improve the performance in the downstream analysis task, and in fact are unable to match sphering (Supplementary Fig. 5). The main reason is that Harmony and GRL require prior knowledge about what the unwanted source of variation is. In high-throughput imaging, the technical

sources of variation are complex and hierarchically organized, and these methods make the assumption that there is a single source of confounding variation organized categorically (batch labels). Sphering is more effective because it does not make such assumptions, and instead uses control cells to model technical variation in an unbiased, non-parametric way.

To further investigate what are the features that characterize treatments or batch effects, we ran a Grad-CAM analysis on a sample of cell images in the BBBC022 dataset. We did not observe any major indication of features that can be localized in the 2D image plane at the single-cell level that could reveal small differences in technical variation (Supplementary Note 2, Supplementary Fig. 6). Batch effects appear more prominently after aggregating single-cell features into population level profiles (Supplementary Fig. 7). This suggests that technical variation (such as wells, plates, and batches) may be accumulated during the aggregation steps while single-cell heterogeneity may be smoothed out. Thus, correcting batch effects and separating biological from technical variation may be more efficient when considering information at various resolutions jointly (single-cells, image-level, well-level, treatment-level).

## Discussion

This paper presents an improved methodology for learning representations of phenotypes in imaging experiments, which uses weakly supervised learning, batch correction with sphering, and an evaluation framework to assess performance. We used this methodology to analyze three publicly available Cell Painting datasets with thousands of perturbations, and we found that: 1) CNNs capture confounding and phenotypic variation as latent variables, 2) the performance of CNNs can be improved by training with datasets that maximize technical and

biological diversity, and 3) batch correction is necessary to recover a representation of the phenotypic outcome of treatments. The WSL approach in our methodology aims to learn *unbiased* features of cellular morphology that can be used to approach various problems and applications in cell biology. This is in contrast to supervised strategies that aim to solve one task with high accuracy, and therefore only capture features relevant to that task. Our approach can also be generalized to other imaging assays or screens, and we anticipate that the same principles will be useful for improving performance in downstream tasks.

We trained a Cell Painting CNN model that can extract single-cell features to create image-based profiles for estimating the phenotypic outcome of treatments in perturbation experiments. Following insights derived from the analysis with our methodology, we constructed a large training resource by combining five sources of Cell Painting data to maximize phenotypic and technical variation for training a reusable feature extraction model. This model successfully improved performance in all three benchmarks while also being computationally efficient (Supplementary Note 3, Supplementary Fig. 8). The fact that the best-performing strategy involved training a single model once and profiling all the three benchmarks without re-training or fine-tuning has a remarkable implication: it indicates that generating large experimental datasets with a diversity of phenotypic impacts could be used to create a single model for the community that could be transformational in the same way that models trained on ImageNet have enabled transfer learning on natural image tasks.

Many machine learning applications typically aim to replicate human behavior (e.g., classifying cats) with 100% accuracy. However, the purpose of image-based profiling is not to mimic human behavior, but rather uncover new knowledge. The accuracy metrics reported in our study (Fig. 5 and Supplementary Fig. 9) are limited by currently available knowledge about these treatments. However, ground truth annotations may be incomplete; uncovering and understanding the mechanisms of these drugs is one of the goals of these studies. The metrics serve the purpose of estimating methodological improvement that can result in future discoveries. Improvements in performance in a drug discovery project could potentially mean new candidate treatments useful for certain diseases.

We used a causal conceptual framework for analyzing the results, which we found very useful to interpret performance differences between feature extraction models. In practice, the framework was useful for guiding decision making while training models, and it is helpful to understand and communicate the challenges of learning representations in imaging experiments. In theory, it also opens new possibilities to formulate the problem in novel ways, for instance, creating learning models that account for all four variables simultaneously. This framework is a compact way to express the causal assumptions of the underlying biological experiment, which is consistent with the experimental evidence that we observed throughout this study. We believe that this is a first step towards studying the causal relationships between disease and treatments using high-throughput imaging experiments and modern machine learning.

Quantifying subtle phenotypic effects remains an ongoing challenge. We successfully trained a model that improved performance after using treatments with strong cellular response, which helped reduce the impact of technical variation on downstream tasks. However, this still does not recover the phenotypic signal of all treatments, and the list of treatments with weak effects remains long. The question whether such effects are overpowered by unwanted variation or if they can even be detected by imaging is still open. Our work presents a strategy to learn representations under real world conditions, and a benchmark to continue study subtle phenotypic effects under noisy conditions.

There are many sources of confounding factors in biological experiments, and microscopy imaging is not exempt. Imaging is a

## Table 3 | Assumptions for variables of the causal graph

| Variable | Type | Encoding | Cardinality | Order |
|---|---|---|---|---|
| Treatments T | Categorical | One-hot vector | $T \in \{0,1\}^2$ | $N \sim O(10^3)$ |
| Images O | Two dimensional | Continuous pixel intensities in a matrix | $O \in R^{W \times H}$ | $W \times H \sim 10^4$ |
| Phenotypes Y | Latent variable | Continuous multi-dimensional vector | $Y \in R^M$ | $M \sim O(10^2)$ |
| Batches C | Categorical | One-hot vector | $T \in \{0,1\}^K$ | $K \sim O(10^2)$ |

powerful technology for observing cellular states, and it is sensitive to phenotypes as well as unwanted variation. If left unaccounted for, unwanted variation can result in biased models that confound biological conclusions, and this is especially true for large capacity, deep learning models. Our experiments show that deep learning can exploit confounding factors to minimize training error, and that batch effect correction is critical to recover the biological representation of interest in conventional or deep-learning based features. This is an active research area with novel solutions being explored, including multi-layer normalization strategies[41], and the use of generative models[42]. Other potential solutions may include invariant risk minimization games[43] or similar formulations.

Deep learning for high-throughput imaging promises to realize the potential of perturbation studies for decoding and understanding the phenotypic effects of treatments. The public release of the JUMP-Cell Painting Consortium's dataset of more than 100,000 chemical and genetic perturbations, collected across 12 different laboratories in academia and industry, is an excellent example of the scale and biological diversity that imaging can bring for drug discovery and functional genomics research[44]. Our Cell Painting CNN is a publicly available model trained specifically for phenotypic feature extraction in image-based profiling studies, and can generalize to new data with new perturbations. We expect that our methodology, together with larger datasets, will be useful to create better models for analyzing images of cells in the future.

## Methods

### Ethical statement

This study does not involve human subjects and does not involve animals or wet-lab experiments. All the relevant ethical principles that apply to data science and machine learning research were observed in this work.

### Causal assumptions

The causal graph in our framework includes four variables: interventions (treatments T), observations (images O), outcomes (phenotypes Y) and confounders (e.g. batches C). Each of the nodes in the graph is a random variable with an associated probability distribution, which is unknown. Table 3 summarizes the assumptions made in practice for each variable.

The exact cardinality of each variable depends on the dataset of focus. A dataset in our study is a particular instance of the graph with a total of P images observed. Therefore, a dataset can be thought of as a matrix with P rows and (N + WxH + M + K) columns, if all the variables were concatenated. In practice, each submatrix is processed by a separate transformation function, notably images are the input to a neural network that predicts the treatments and recovers the phenotype in a hidden feature layer (see Weakly Supervised Learning below for more details).

Note that confounders involve a wide range of technical and unwanted variation, not limited to batch effects only. There are also well-position effects, and plate effects, among others, which are hierarchically organized sources of variation that typically confound the

analysis. In the literature, all these sources of variation are usually grouped and called batch effects, because addressing them separately is still an open research problem. In our work, we do not make explicit distinction of what is the source of the confounding factors, and for that reason, we model it as an unobserved latent variable.

For simplicity, we assume that there is a single context (**X**, not in the diagram) for experimental treatments, which are clonal cells of an isogenic cell line; perturbation experiments with multiple cell lines may need a different model. We assume that images and treatments are observables (**O** and **T**) because the images are acquired as a result of the experiment, and the treatments are chosen by researchers. We also assume that the phenotype and confounders are latent variables (**Y** and **C**) that we want to estimate and separate.

The relationships in the causal graph are interpreted as follows: the arrow from **T** to **Y** indicates that treatments are applied to cells and are the main direct biological cause of phenotypic changes in cells in the perturbation experiment. The arrow from **Y** to **O** indicates that we partially observe the phenotypic outcome through images. This observation is assumed to be noisy and incomplete, requiring hundreds of cells and multiple replicates to increase the chances of measuring the real effect of treatments. In addition, the image acquisition process and the overall experiment are influenced by technical variation. The arrow from **C** to **O** indicates that images are impacted by artifacts in image acquisition, including microscope settings and assay preparation. The arrow from **C** to **Y** indicates that phenotypes are impacted by variations in cell density and other conditions that make cells grow and respond differently. The arrow from **C** to **T** indicates that treatments are impacted by plate map designs that are not fully randomized and usually group treatments in specific plate positions.

We observe treatment outcomes (**Y**) indirectly through imaging assays, and thus, we need image analysis to recover the phenotypic effect and to separate it from unwanted variation (**C**). A representation of the phenotypic effect can be obtained with the workflow depicted in Fig. 1C-E, which illustrates three major steps: 1) modeling the correlations between images and treatments using a CNN trained with weakly supervised learning (WSL), 2) using batch correction to learn a transformation of the latent representation of images obtained from intermediate layers of the CNN, and 3) generating representations of treatment effects in cellular morphology for downstream analysis.

## Compound concentration and phenotypic strength

There is an important distinction between compound concentration and phenotypic strength. Although they are causally related, they are different concepts. First, the concentration of compounds was predetermined as part of the experimental design. Two of our three evaluation datasets (BBBC022 and BBBC036) are compound screens, and the third dataset is a gene overexpression experiment at a single "dosage" (BBBC037). The compound screens prioritized diversity of compounds with a fixed dose instead of multiple concentrations for titration studies. Therefore, for most of the compounds in BBBC022 and BBBC036 only a single concentration was available, usually a high one. Second, the strength of phenotypes is the effect that perturbations produce on cells, regardless of their type or concentration. A high dose does not mean that the phenotypic effect is strong, because some compounds may not induce a phenotypic effect that can be detected with imaging, even at high concentrations. Therefore, it is important to separate the cause (compound dose) from the effect (phenotypic change) to correctly interpret the results.

## Average treatment effect

To estimate Average Treatment Effect (ATE) we use the Euclidean distance between feature representations obtained with classical features as an independent measure of cellular morphology. The Euclidean distance is a useful estimator of ATE, because it is by definition the expected difference between two outcomes. We approximate this quantity with the Euclidean distance between negative controls and treatments given that profiles are high-dimensional and we aim to capture the total effect (distance). The cosine similarity was not used in this context because it only measures the directionality of the effect and ignores its magnitude. We consistently use the cosine distance among treatment profiles in all the biological matching tasks involving MoAs (see Methods Similarity Matching for more details).

## Weakly supervised learning

Weakly supervised learning[23] (WSL) trains models with the auxiliary task of learning to recognize the treatment applied to single cells. Treatments are always known in a perturbation experiment, while other biological annotations, such as mechanism of action or genetic pathway may not be known for certain treatments, only reflect part of the phenotypic outcome, and is usually unknown at the single-cell level (which is the resolution used for training models in this work). We use an EfficientNet[45,46] architecture with a classification loss with respect to treatment labels for training WSL models.

WSL captures the correlations between observed images and treatments and makes the following assumptions: 1) if a treatment has an observable effect then it can be seen in images, and therefore, training a CNN helps identify visual features that make it detectably different from all other treatments. 2) Treatment labels in the classification task are weak labels because they are not the final outcome of interest, they do not reflect expert biological ground truth, and there is no certainty that all treatments produce a phenotypic outcome, nor that they produce a different phenotypic outcome from each other. 3) Cells might not respond uniformly to particular treatments[47], which yields heterogeneous subpopulations of cells that may not be consistent with the treatment label, i.e, treatment labels do not have single-cell resolution. 4) Intermediate layers of the CNN trained with treatment labels capture all visual variation of images as latent variables, including confounders and causal phenotypic features.

## Image preprocessing

The original Cell Painting images in all the datasets used in this work are encoded and stored in 16-bit TIFF format. To facilitate image loading from disk to memory during training of deep learning models, we used image compression. This is only required for training, which requires repeated randomized loading of images for minibatch-based optimization.

The compression procedure is as follows:
Compute one illumination correction function for each channel-plate[48]. The illumination correction function is computed at 25% of the width/height of the original images.
Apply the illumination correction function to images before any of the following compression steps.
Stretch the histogram of intensities of each image by removing pixels that are too dark or too bright (below 0.05 and above 99.95 percentiles). This expands the bin ranges when changing pixel depth and prevents having dark images as a result of saturated pixels.
Change the pixel depth from 16 bits to 8 bits. This results in 2X compression.
Save the resulting image in PNG lossless format. This results in approximately 3X compression.
The resulting compression factor is approximately (2X)(3X) = 6X.

This preprocessing pipeline is implemented in DeepProfiler and can be run with a metadata file that lists the images in the dataset that require compression, together with plate, well and site (image or field of view) information. Note that compression is only used to optimize storage space, data transferring and computing time, especially given the large size of uncompressed datasets (which can be in the order of TBs). Histogram clipping is not meant to enhance image contents, it is used as a preprocessing step before compressing pixel depth from 16

bits (64 K pixel values) down to 8 bits (256 pixel values). Histogram clipping reduces the impact of noise in the tails of the original pixel distribution and to prevent allocation of compressed bits on unnecessarily high or low levels of brightness. The percentiles are relative to each individual image.

## EfficientNet

The deep convolutional neural network architecture used in all our experiments is the EfficientNet[45,46]. We use the base model EfficientNet-B0 to compute features on single-cell crops of 128 × 128 pixels. It consists of 9 stages: input, seven inverted residual convolutional blocks from MobileNetV2[49] (with the addition of squeeze and excitation optimization) and final layers. The usage of convolutional blocks from MobileNetV2 in combination with neural architecture search gave EfficientNet an advantage in terms of computational efficiency and accuracy compared to ResNet50. This model has only 4 million parameters and can extract features from single cells in a few milliseconds using GPU acceleration. EfficientNet has been previously used for image-based profiling, including in models trained with the CytoImageNet dataset[50], by top competitors in the Recursion Cellular Image Classification challenge in Kaggle, and for studying variants of unknown significance in cancer lung cells[10].

## Training cell painting models

The Cell Painting models are trained on single-cell crops obtained from full images using cell locations and full-image metadata. Cropped single cells are exported to individual images together with their segmentation mask if available. In all our experiments, we used single cells cropped from a region of 128 × 128 pixels centered on the cell's nucleus without any resizing. We preserve the context of the single cell (background or parts of other cells), meaning that the segmentation mask is not used in our experiments.

To train a weakly supervised model we first initialize an EfficientNet with ImageNet weights and sample mini-batches of 32 examples for training with an SGD optimizer with a learning rate of 0.005. We train models for 30 epochs; each epoch makes a pass over example cells of all treatments in a balanced way. Balancing is set to draw the same number of single cells from each treatment (the median across treatments) in one epoch, and every epoch resamples new cells from the pool. This strategy leverages the variation of cells in treatments that are overrepresented (such as controls), and oversamples cells from treatments with fewer than the median. Balancing is important to optimize the categorical cross-entropy loss to compensate for rare classes among the hundreds of treatments used for training in our experiments.

All single cells go through a data augmentation process during training, which involves the following three steps:

1. Random crop and resize. This augmentation is applied with 0.5 probability. The crop region size is random, 80% to 100% of the size of the original image, then resized back to 128 × 128 pixels.
2. Random horizontal flips and 90-degree rotations.
3. Random brightness and contrast adjustments, each channel is augmented and renormalized separately.

Data augmentation is a strategy for regularization; it is used to prevent models from learning simple associations and to force the model to find alternative explanations. Crop and resize may change cell size, which is a biologically meaningful feature. However, it is not the only feature that we aim to capture in Cell Painting studies. Other simpler assays and algorithms can be used for that purpose. Instead, Cell Painting captures a wider range of structural and morphological cellular variations that are subtle and important for distinguishing the effects of thousands of perturbations. In practice, we observed that random cropping and resizing improves training performance and prevents overfitting. This is because treatments that impact cell size

also tend to impact other features as a result, including colocalization of stains and textures. These features are then captured robustly when random variations in cell size are introduced as an augmentation during training.

In addition, introducing random brightness and contrast augmentations prevents overfitting to non-biological illumination variations, resulting in more robust morphological features. The key observation is that pixel brightness is meaningful relative to other structural or morphological patterns in the same channel, but not in absolute values. By randomly changing the absolute brightness and contrast of each channel individually, we simulate technical artifacts that could result in one of the channels having abnormal illumination variation (either brighter or darker than usual). Note that all the pixels in the same channel change by the same amount. This facilitates learning features that describe structural patterns in the image regardless of potentially unexpected illumination changes.

Instead of data augmentations, equivariance can be used as part of the model, which has been investigated in other domains, including pathology. We leave this possibility for future research.

We used two data-split approaches for creating training and validation subsets for the single-cell treatment classification task:

Leave plates out: we selected the plates in a way that the data of one subset of plates is only the train data and another one is in validation.

Leave cells out: all plates are used for training and validation. We randomly choose single cells from each well, meaning that approximately 60% of cells from each well would be in the training set and the rest in the validation set.

The training of deep learning models was performed on NVIDIA DGX with NVIDIA V100 GPUs and servers with NVIDIA A6000 GPUs; a single GPU was used to train each model.

## Feature extraction with trained Cell Painting models

We extract features of single cells and store them in one NumPy file per field of view using an array of vectors (one per cell). The feature extraction procedure requires access to full images, metadata and location files. Since a trained model is natively trained for five-channel images, there is no need to replicate each channel separately, thus, each cell requires one inference pass and the feature vector contains representation of all channels simultaneously. The size of the feature vectors is 672 for reported results, which were extracted from EfficientNet's block6a-activation layer.

## ImageNet pre-trained models

ImageNet pre-trained convolutional networks are widely used in computer vision applications and they have also been used in image-based profiling applications. ImageNet is a large collection of natural images with objects and animals of different categories[51]. A deep learning model trained on this dataset is capable of extracting generic visual features from images for different applications, also known as transfer learning. Pre-trained models for morphological profiling have been evaluated in several studies[10,11,17,18,33,52,53].

We use DeepProfiler to extract features of single-cells with the pre-trained EfficientNet-B0 model available in the Keras library. The size of single-cell images in our experiments is 128 × 128 after being cropped from field-of-view images. An ImageNet pre-trained model expects images of higher resolution, specifically 224 × 224 in our case; therefore the cell crops are first resized. The pixel values are then rescaled using min-max normalization and adjusted to have values [−1:1] to match the required input range. As Cell Painting images are five-channel and ImageNet pre-trained models expect three-channel (RGB) images, we follow the well established practice of computing a pseudo RGB image for each grayscale fluorescent channel by replicating it three times before passing it through the model (thus each

cell requires five inference passes). Features extracted for each channel are concatenated and the resulting feature vector size is 3,360 (672 is the size of the block6a-activation layer of the EfficientNet used in our experiments).

## Multiple instance learning with attention

Multiple instance learning (MIL) is a learning technique for datasets with weak labels, where the training examples are assorted into sets of examples – bags, and each bag has a single label[54]. In the binary classification setting, the bag is assigned as negative if all examples in the bag are negative, otherwise, the bag is assigned as positive. This technique, paired with an attention mechanism, was used to analyze histology images[32] where the prediction of the bag would stand for the diagnosis (given that examples come from the same sample).

We implemented this approach as an alternative weakly supervised learning method. Given the multi-class setting, the construction of bags was adjusted as follows: 1) bag size is 16, 2) the bag is labeled as negative control class (DMSO or EMPTY) if all cells in the bag come from negative control wells, 3) bags are labeled with a treatment class when it contains examples of negative controls mixed with examples of the corresponding treatment. 4) Treated bags have 4 to 12 examples (25-75%) belonging to the treatment class and the rest are randomly sampled from negative controls.

To train a weakly supervised model with MIL we first initialize an EfficientNet with ImageNet weights and batch size of 32 bags, then training with an SGD optimizer with a learning rate of 0.005. We train models for 50 epochs; each epoch makes a model pass over bags of all treatments in a balanced way. Bags are resampled at every epoch, and for each treatment class we generate the same number of bags. The augmentation pipeline was the same as for the training of other Cell Painting models and the MIL model was trained only with the combined Cell Painting dataset using leave-cells-out data split (see Methods).

## Online label smoothing

Online label smoothing (OLS)[31] is a technique for training models with weak or noisy labels. During the training process the labels are dynamically updated at the end of each epoch. The update rule for label $L$ of the training example is the following:

$$L = \alpha L_{hard} + (1 - \alpha)L_{soft} \tag{1}$$

where $\alpha$ sets the shift from the current label to the new predicted label. At the beginning of training, $L_{hard}$ is the ground-truth label and later, it is the label computed in the previous epoch. $L_{soft}$ is the predicted label after the training epoch is finished. We used OLS to train a model with the combined Cell Painting dataset using the leave-cells-out validation scheme (see Methods). The parameter $\alpha$ for the label update rule was set to 0.03. Additionally, the label smoothing[55] parameter was set to 0.1.

## Gradient reversal layer

Gradient reversal layer (GRL)[40] is an approach to domain adaptation for representation learning. It works by adding an adversarial classifier in a separate network branch that classifies the known domains of data points. The gradient calculated in this branch is used to move the parameters of the model in the opposite direction of the classifier's best performance, thus preventing the model from learning solutions that depend on the domain information.

The training setup with gradient reversal layer does not differ from training other Cell Painting models (see Methods), except for the addition of a classifier branch for domain classification and gradient reversal layer. The branch for domain classification consisted of three dense layers (of sizes 1024, 512 and 128) and the softmax function for class prediction. We trained WSL + GRL for each benchmark dataset

separately and considered experimental plates as domains, which means that the number of domains for each dataset was different (5 for BBBC037, 20 for BBBC022 and 55 for BBBC036).

## Segmentation

The cell segmentation for the benchmark datasets (BBBC037, BBBC022 and BBBC036) was performed with methods built in Cell-Profiler v2 based on Otsu thresholding[56] and propagation method[57] based on Voronoi diagrams[58] or watershed from[59]. The segmentation is two-stepped: first, the images stained with Hoechst (DNA channel) were segmented using global Otsu thresholding. This prior information is then used in the second step: cell segmentation with the propagation or watershed method. The input channel for the second step depends on the dataset, as well as the other specific parameters of segmentation. The segmentation part of the pipelines is available in the published CellProfiler pipelines (see Code availability section). For the purposes of this project, we used the center of the nuclei to crop out cells in a region of 128 × 128 pixels. These cell crops are used in all the deep learning workflows.

## Feature extraction with CellProfiler

CellProfiler[60] allows the construction of customizable automated pipelines for analysis of biological images. It facilitates the analysis and extraction of meaningful information from high-throughput imaging experiments. The pipeline starts with image import, then it is followed by image pre-processing, such as illumination correction and noise removal. Then the objects are identified: nuclei are identified first and with this prior information the boundaries of the whole cell are inferred. Once cells are segmented, CellProfiler extracts feature vectors per cell, which are designed to be human readable and grouped by cell region (nucleus, cytoplasm or cell). Each of those feature groups has several common subgroups, such as shape features, intensity-based features, texture features and context features. Table 4 lists those feature subgroups.

Feature extraction for the evaluated datasets was performed with CellProfiler version 2. The table above lists the groups and names of features used in our experiments. Most of these features are extracted from each channel independently, except for the correlation features. The resulting size of single-cell feature vectors is approximately 1,700 (the exact value can vary among datasets). In our analysis, we used well-level aggregated profiles (see also "Feature aggregation and profiling") to obtain baseline results. We reused publicly available features that were computed by the authors of the original study (BBBC037[61], BBBC022[37] and BBBC036[38]). The links to the original data sources are listed in the Data Availability section below.

## Feature aggregation and profiling

Image-based profiling aims to create representations of treatment effects, which is obtained by aggregating information of single cells into population-level profiles. This process follows a multi-step aggregation process. Features of single-cells are first aggregated using the median operator at field-of-view (image) level. Next, fields-of-view features are aggregated using the mean to create a well-level profile. Finally, treatment-level profiles are obtained with the average across replicate wells. The feature aggregation steps are the same for CellProfiler and deep learning features. CellProfiler well-level features with NA values were removed in the aggregation pipeline.

This choice of alternating between median and mean follows the fact that at the field-of-view level there is more data (hundreds of cells), which is potentially noisy and prone to errors (from segmentation or feature extraction). Therefore the median is used as the robust estimator of the local morphological trend. At the replicate level, there are only less than ten data points to aggregate (9 fields of view, or 5 replicates), and then the average is computed to capture a smooth

**Table 4 | List of CellProfiler features used for profiling cellular morphology**

| Feature Group | Features and description |
|---|---|
| AreaShape | Compactness, Eccentricity, Extent, FormFactor, MajorAxisLength, MaxFeretDiameter, MaximumRadius, Mean-Radius, MedianRadius, MinFeretDiameter, MinorAxisLength, Orientation, Perimeter, Solidity, Zernike shape features. |
| Correlation, Correlation_Manders, Correlation_RWC | Those correlations are calculated for pairs of channels. |
| Granularity | Granularity for each channel and each instance of the granularity spectrum. |
| Intensity | IntegratedIntensityEdge, LowerQuartileIntensity, MADIntensity, MassDisplacement, MaxIntensityEdge, Mean-Intensity, MedianIntensity, MinIntensity, StdIntensityEdge, StdIntensity, UpperQuartileIntensity. Each feature is calculated per channel. |
| Location | CenterMassIntensity, MaxIntensity. Each location is calculated per channel. |
| Neighbors | AngleBetweenNeighbors, FirstClosestDistance, NumberOfNeighbors, PercentTouching, SecondClosestObjectNumber. Features in this group are only calculated for "Cell" and "Nuclei" groups. |
| RadialDistribution | FracAtD, MeanFrac, RadialCV. Each feature was calculated per channel. |
| Texture | AngularSecondMoment, Contrast, Correlation, DifferenceEntropy, DifferenceVariance, Entropy, Gabor, InfoMeas1\2, InverseDifferenceMoment, SumAverage, SumEntropy, SumVariance, Variance. Each feature was calculated per channel. |

trend of morphological features among them. This pipeline was originally proposed by[62], and has been used in many other studies[18].

## Batch correction with the sphering transform

To recover the phenotypic features of treatments from the latent representations of the weakly supervised CNN, we employ a batch correction model inspired by the Typical Variation Normalization (TVN) technique[18]. This transform aims to reduce the variation associated with confounders and amplify features caused by phenotypic outcomes (Fig. 1D). The main idea of this approach is to use negative control samples as a model of unwanted variation under the assumption that their phenotypic features should be neutral, and therefore differences in control images reflect mainly confounding factors. We follow this assumption and use a sphering transformation to learn a function that projects latent features from the CNN to a corrected feature space that preserves the phenotypic features caused by treatments. We note that given how control wells are placed in plates, they may not represent all of the unwanted variation caused by plate layout effects, nevertheless, we assume it is a sufficient approximation.

In our implementation, we aim to reduce the profiles of control wells to a white noise distribution using a sphering transform, and then use the resulting transformation as a correction function for treated wells. First, the orthogonal directions of maximal variance are identified using singular value decomposition (SVD) on the matrix of control wells. Then, directions with small variation are amplified while directions with large variation are reduced by inverting their eigenvalues. We control the strength of signal amplification or reduction with a regularization parameter. The computation involves only profiles of negative controls and as a result, we obtain a linear transformation that can be applied to all well-level feature vectors in a dataset.

The sphering transformation takes $n$ well-level profiles of negative controls with vector size $d$ as an input matrix $X^{n \times d}$. Then, its covariance matrix is calculated as $\Sigma = \frac{X^T X}{n}$ followed by eigendecomposition $\Sigma = U \Delta U^T$, where $\Delta$ is the diagonal matrix of eigenvalues, and $U$ is the matrix of orthonormal vectors. We renormalize the orthonormal vectors by inverting the square root of the eigenvalues in $\Delta$ together with a regularization parameter $\lambda$. The resulting ZCA-transformation[63] matrix is $Q = U(\Delta + \lambda)^{-\frac{1}{2}} U^T$, which can be used to compute the corrected profile of a treated well $t$ with a matrix multiplication: $t' = Qt$. The effect of sphering and its regularization on representations and profiling performance is presented in Fig. 4 and Supplementary Fig. 5. This transformation was originally studied for image-based profiling in detail by Michael Ando and others[18], and has been subsequently used in many other studies[5,9,10,23,64–69]. Our implementation is a simplified version of TVN following the principles of the sphering transform[63,70,71].

The selection of the regularization parameter $\lambda$ is critical for obtaining improved results as presented in Fig. 6. In our work, we select this parameter by inspecting the plot and identifying a value that maximizes quantitative performance for each feature extraction strategy independently. When performance is saturated, we select the largest parameter that provides stable results. In practice, we found $10^{-3}$ to be a good choice in many cases.

## Harmony

Harmony[39] was originally developed to integrate single-cell RNA-seq and spatial transcriptomics data from various datasets. The input data is projected into lower dimension space with PCA and then clustered with a modified k-means algorithm. Harmony iteratively adjusts single data points (cells) with a series of corrections inferred with the help of clustering, until the final clustering is stable.

In our experiments, we used Harmony on its own and combined with sphering transform, when sphering is applied first and then Harmony (in both cases on well-level profiles). Experimental plates were treated as "datasets" and wells as "cells". For the number of clusters parameter we used 205 in BBBC037 (one per perturbation), and 300 for BBBC022 and BBBC036. Applying Harmony slightly improves performance in the biological matching task if sphering was not used in all metrics\datasets, but it is not consistent when combined with sphering (Supplementary Fig. 5).

## Similarity matching

To assess the similarity between treatment profiles the cosine similarity is measured between pairs of treatments. The cosine similarity is one of several similarity metrics that can be used in profiling[12] and has been used in previous studies[62].

$$cosine\ similarity = \frac{A \cdot B}{||A||\ ||B||} \qquad (2)$$

where A and B are image-based profiles, i.e., multidimensional vectors. We adopt the cosine similarity in all our similarity search and biological matching experiments.

## Evaluation metrics

For quantitative comparison of multiple feature extraction strategies, we simulate a user searching a reference library to find a "match" to their query treatment of interest. We used a leave-one-treatment-out strategy for all annotated treatments in three benchmark datasets, following previous research in the field[18,61,62]. In all cases, queries and

reference items are aggregated treatment-level profiles matched using the cosine similarity (Methods). The result of searching the library with one treatment query is a ranked list of treatments in descending order of relevance. A result in the ranked list is considered a positive hit if it shares at least one biological annotation in common with the query; otherwise it is a negative result (Fig. 1G).

There are several quantitative evaluations of feature representation quality that we use in our study. At the single-cell level, we expect neural networks to classify single cells into their corresponding treatment, and therefore use accuracy, precision and recall to evaluate performance (see Fig. 3 and main text). For downstream analysis we adopted a biological matching task, which simulates a user searching for treatments that correspond to the same mechanism of action or genetic pathway (for compound and gene overexpression perturbations respectively). These queries are conducted and evaluated at the treatment-level, and the main idea is to assess how well connected treatments are in the feature space according to known biology.

As ground truth annotations for evaluating downstream biological tasks, we used mechanism of action (MoA) labels publicly available in compound libraries. These annotations may not be 100% accurate for several reasons; for example, the phenotype of compounds may vary depending on the tested dose resulting in a different MoA for the same compound. The MoA annotations have not been manually and individually confirmed for each compound in the study; they only represent a potentially expected phenotype according to what we know in the literature about these compounds. Factors such as the dosage, the sensitivity of imaging, and confounders, among others may determine whether the MoA association is correct or not. The quantitative evaluation presented in this study is an attempt to measure how likely two related phenotypes are given the observations and the available annotations.

We use two main metrics for evaluating the quality of the results for a given query: 1) folds of enrichment and 2) mean average precision (mAP). The folds-of-enrichment metric (see details below) is inspired by statistical analyses in biology and determines how unusual positive connections happen to be in the top 1% of the list[61]. On the other hand, the mAP metric is inspired by information retrieval research, and quantifies the precision and recall trend over the entire list of results for all queries.

In order to simulate queries, we proceed as follows:
Choose a query treatment - which belongs to an MoA or pathway that has at least two treatments in the database and, therefore, it is possible to find a match.
Library treatments - all the others while leaving the query treatment out. Library treatments represent a database of treatments with known MoAs or pathways annotations, which can be candidate matches for a given query.

## Folds of enrichment

For each query treatment we calculate the odds ratio of a one-sided Fisher's exact test. The test is calculated using a $2 \times 2$ contingency table: the first row contains a number of treatments with the same MoAs or pathways (positive matches) and different MoAs or pathways (negative matches) at a selected threshold of the list of results. The second row is the same, but for the treatments below the threshold (the rest). Odds ratio is a sum of the first row divided by the sum of the second row. It estimates the likelihood of observing the treatment with the same MoA or pathway in the top connections.

We calculate the odds ratio of each individual query, and then obtain the average over all query treatments. The threshold we use is 1% of connections, meaning we expect the top 1% of matching results in the list to be significantly enriched for positive matches. This metric in the text is referred to as "Folds of Enrichment". The implementation of the metric is available as a part of analysis pipelines (see Code availability section).

## Mean average precision

For each query treatment, average precision (area under precision-recall curve) is computed following the common practice in information retrieval tasks. The evaluation starts from the most similar treatments to the query (top results) and continues until all positive pairs (response treatments with the same MoA or pathway) are found. Precision and recall are computed at each item of the result list.

$$Precision = \frac{TP}{TP + FP} \qquad (3)$$

$$Recall = \frac{TP}{TP + FN} \qquad (4)$$

where TP are the true positives, FP are the false positives, and FN are the false negatives in the list of results until the current item. This is evaluated for each query separately. As the number of treatments per MoA or pathway is not balanced, the precision-recall curve has a different number of recall points. Therefore, precision and recall are interpolated for each query to cover the maximum number of recall points possible in the dataset, and thus allow for averaging at the same recall points. The interpolated precision at each recall point is defined as follows[72]:

$$p_{inter}(r) = \max_{r' \geq r} p(r') \qquad (5)$$

Average precision for a query treatment is the mean of $p_{inter}$ at all recall points. The reported mean average precision (mAP) is the mean average precision over all queries.

## Benchmarks and ground truth annotations

For this study, we used three publicly available Cell Painting datasets representing gene overexpression perturbations (BBBC037[61]) and compound perturbations (BBBC022[37] and BBBC036[38]). The three datasets were produced at the Broad Institute using the U2OS cell-line (bone cancer) following the standardized Cell Painting protocol[1], which stains cells with six fluorescent dyes and acquires imaging samples in five channels at 20X magnification. The compound perturbation experiments used DMSO as a negative control treatment, while in the gene overexpression experiments no perturbation was used for negative control samples. All experiments were conducted using multiple 384-well plates at high-throughput with 5 replicates per treatment (except for high-replicate positive and negative controls).

We conducted quality control of images in all the three datasets by analyzing image-based features with principal component analysis. The outliers observed in the first two principal components were flagged as candidates for exclusion, and were visually inspected to confirm rejection. We found most of these images to be noisy or empty and not suitable for training and evaluation. With this quality control, two wells were removed from BBBC037, 43 wells from BBBC022, and no wells were removed from BBBC036. If treatments had multiple concentrations in BBBC022 and BBBC036, we kept only the maximum concentration for further analysis and evaluation.

The ground-truth annotations for compounds correspond to mechanism-of-action (MoA) labels when known, and can include multiple annotations per compound. In the case of gene overexpression, the ground-truth corresponds to the genetic pathway of perturbed genes. We used annotations collected in prior work for the same three datasets[61] and applied minor updates and corrections (see Data Availability section). Only treatments with at least two replicates left after quality control are included in the ground-truth.

## Measuring treatment effect

The effect of treatments is approximated by computing the distance between the morphological features of treatments and controls. We

use batch-corrected features obtained with CellProfiler (with sphering regularization parameter 1e-2) in the following way:

1. Calculate the median profile of control wells within the same plate (median control profile of the plate).
2. For each treated well, calculate the Euclidean distance between its well-level profile and the median control profile of its plate.
3. Estimate the distribution of control well distances against the median control profile per plate. Then, calculate their mean and standard deviation.
4. Using the statistics of control distances per plate, Z-score the distances of treated wells obtained in step 2.
5. Finally, we define the approximate measure of the effect of a given treatment as the average of the Z-scores of its well replicates across plates.

Intuitively, we expect treatments with stronger effects to have a high average Z-score while treatments with weaker or no detectable effect are expected to have low average Z-score. We use this measure to rank treatments and select subsets of treatments for evaluation of the impact of treatment effect during training, as well as for sampling treatments with high phenotypic effect for creating a combined training dataset.

## Combined Cell Painting dataset

We combined five publicly available Cell Painting datasets to create a training resource that maximizes both phenotypic and technical variation. The five dataset sources include the three benchmarks described above (BBBC037, BBBC022, and BBBC036), as well as two additional datasets: 1) BBBC043[10], a gene overexpression experiment to study the impact of cancer variants, and 2) LINCS[5], a chemical screen of FDA approved compounds for drug repurposing research. Both BBBC043 and LINCS are perturbation experiments conducted with A549 cells (lung adenocarcinoma). In total, these five sources of data have more than 6,000 treatments, in hundreds of plates, thousands of wells, and millions of images resulting in the order of hundreds of millions of imaged single cells. Our goal was to select a sample of single cells from these five sources to maximally capture phenotypic and technical variation.

Instead of sampling single cells uniformly at random, we follow the distribution of treatments to include biological diversity, and the organization of the experimental design to represent various sources of technical noise. Technical variation is organized hierarchically in experiments, starting with the five sources of Cell Painting images, continuing with batches, plate-maps, plates, and well positions. We aimed to bring samples from as many of these combinations as possible to have cells representing different types of technical variation. In terms of biological variation, three of the five data sources have U2OS cells and the other two have A549 cells, resulting in two different cellular contexts being represented. The five sources also include two types of perturbations (chemical and genetic), and multiple treatments.

To preserve as much phenotypic variation as possible, we sample treatments from both cell lines, both types of perturbations, and we identify the treatments with strongest effect in each of the five sources following the procedure described in the previous section. Several treatments overlap across data sources, and we prioritized those that can be found in two or more sources simultaneously. An example is negative controls: all compound screens use DMSO as the negative control, and we would expect their phenotype to match across data sources. The same expectation holds for the rest of treatments. Negative control wells are typically present in each plate of the experiment in several replicates.

The selection of strong treatments started with the BBBC022 and BBBC036 datasets (chemical perturbations). We selected the

500 strongest treatments (see Measuring treatment effect) from BBBC022 and searched for those in BBBC036, which resulted in 301 strong treatments in common between both datasets. We additionally selected 50 unique treatments from BBBC022 and 62 unique treatments from BBBC036. Out of those 413 treatments, 122 overlapped with the LINCS dataset and were included. We additionally selected 7 random treatments from LINCS, from top 20 (by number of associated treatments) MoAs. Treatment selection from BBBC037 and BBBC043 (gene overexpression perturbations) was similar, and we identified 28 overlapping genes. We assume that "wildtype" genes from both datasets are the same, and then we selected the 29 strongest unique perturbations from the BBBC037 dataset and the 32 strongest perturbations from BBBC043 from non-overlapping subsets.

Negative controls from compound screening datasets and negative controls from gene overexpression datasets are considered as different classes in the combined dataset (DMSO and EMPTY). Not all control wells were included from the LINCS dataset in the final sample, as these would result in extreme overrepresentation, so we randomly sampled three control wells per plate. As the final step, the treatments with less than 100 cells were filtered out. In total the dataset contains 490 classes (488 for treatments and 2 for negative controls), 8,423,455 individual single-cells (47% treatment and 53% control cells). See Venn diagrams in Fig. 4 for more details.

## Statistics and reproducibility

This study uses machine learning to analyze publicly available Cell Painting datasets with the goal of understanding image-based profiling methodologies. No statistical method was used to predetermine sample size. Certain images were excluded from the analysis following the quality control procedure described in the "Benchmarks and ground truth annotations" section of the Methods. The experiments were not randomized, and data training-validation splits were created as described in the "Training Cell Painting models" section of the Methods. The Investigators were not blinded to allocation during experiments and outcome assessment.

## Reporting summary

Further information on research design is available in the Nature Portfolio Reporting Summary linked to this article.

## Data availability

The Cell Painting datasets used in this study (raw images and CellProfiler profiles) are available at public S3 buckets that can be accessed using the AWS Command Line Interface (CLI). Please install the AWS CLI following the instructions on this link: https://docs.aws.amazon.com/cli/latest/userguide/getting-started-install.html. Then, to download data use the "cp" command with the "--recursive" and "--no-sign-request" flags. For example, BBBC037 dataset can be downloaded with the following command: 'aws s3 cp s3://cytodata/datasets/TA-ORF-BBBC037-Rohban/./ --recursive --no-sign-request'. The following are S3 URLs needed to get all the data, including images and precomputed features. **BBBC037** gene overexpression dataset in U2OS cells[36], s3://cytodata/datasets/TA-ORF-BBBC037-Rohban/. **BBBC022** compound screening in U2OS cells[37], s3://cytodata/datasets/Bioactives-BBBC022-Gustafsdottir/. **BBBC036** compound screening in U2OS cells[38], s3://cytodata/datasets/CDRPBIO-BBBC036-Bray/. **BBBC043** gene overexpression dataset in A549 cells[10], s3://cytodata/datasets/LUAD-BBBC043-Caicedo/. **LINCS** compound screening in A549 cells[5], s3://cellpainting-gallery/cpg0004-lincs/broad/images/2016_04_01_a549_48hr_batch1/. **Combined Cell Painting dataset** that was collected in this study is available in Cell Painting Gallery S3 bucket: s3://cellpainting-gallery/cpg0019-moshkov-deepprofiler/. Source data (SourceData.xlsx) is provided with this paper. Source data are provided with this paper.

## Code availability

To run all the experiments in this study, we developed DeepProfiler, a tool for learning and extracting representations from high-throughput microscopy images using convolutional neural networks (CNNs). DeepProfiler uses a standardized workflow that includes image pre-processing, training of CNNs and feature extraction, as discussed in previous sections. DeepProfiler is implemented in Tensorflow[73] (version 2) and is publicly available on GitHub https://github.com/cytomining/DeepProfiler[74]. The documentation of DeepProfiler is available in the following link: https://cytomining.github.io/DeepProfiler-handbook/, and describes the steps for installing, configuring and running the software for profiling new images and for training models. In DeepProfiler we used the following EfficientNet implementation: https://github.com/qubvel/efficientnet. Additionally, DeepProfiler code is available on Zenodo https://doi.org/10.5281/zenodo.10410958[74]. The processing and profiling pipelines for the three benchmarks evaluated in this work (Jupyter notebooks and Python scripts to analyze features) are available on GitHub: https://github.com/broadinstitute/DeepProfilerExperiments and Zenodo. https://doi.org/10.5281/zenodo.10419640[74] This repository also includes the DeepProfiler configuration files used for training the models on each dataset, as well as the configuration for training the Cell Painting CNN model. In addition, the ground truth files and code for evaluation of the downstream tasks are also available in this repository. The Cell Painting CNN model (trained with leave-cells-out training-validation split) is available on Zenodo: https://doi.org/10.5281/zenodo.7114557. The ImageNet pre-trained EfficientNet model used in this study can be found here: https://github.com/Callidior/keras-applications/releases/download/efficientnet/efficientnet-b0_weights_tf_dim_ordering_tf_kernels_autoaugment.h5. The code and CellProfiler pipelines for three evaluated datasets can be found in the following GitHub repositories: BBBC037: https://github.com/carpenterlab/2017_rohban_elife, BBBC036: https://github.com/gigascience/paper-bray2017, and BBBC022: Supplementary materials in ref. 37.

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

## Acknowledgements

We thank Salil Bhate for the valuable discussions and feedback provided to improve the clarity of this manuscript. NM acknowledges the short-term scientific mission grants provided by eCOST action CA15124 (NEUBIAS) in 2019 and 2020. NM and PH acknowledge support from the LENDULET-BIOMAG Grant (2018-342), from the European Regional Development Funds (GINOP-2.2.1-15-2017-00072), from the H2020 and EU-Horizont (ERAPERMED-COMPASS, ERAPERMED-SYMMETRY, DiscovAIR, FAIR-CHARM, SWEEPICS), from TKP2021-EGA09, from the ELKH-Excellence grants and from the Cooperative Doctoral Programme (2020-2021) of the Ministry for Innovation and Technology, from OTKA-SNN no.139455/ARRS. Researchers in the Carpenter–Singh lab were supported by NIH R35 GM122547 to AEC. Researchers in the Cimini lab were supported by NIH P41 GM135019. AG was supported by grant number 2018-192059 and BAC was additionally supported by grant number 2020-225720 from the Chan Zuckerberg Initiative DAF, an advised fund of the Silicon Valley Community Foundation. JCC was supported by the Schmidt Fellowship program of the Broad Institute and by the NSF DBI Award 2348683.

## Author contributions

NM: conceptualization, experiments, data processing, data analysis, software development, software testing, documentation, writing. MB: experiments and data processing. SB: data processing, software development, software testing. MS: data processing, software development, software testing. CM: software development. AG: software development. RAS: software testing, documentation, writing. YH: software testing and documentation. MB: conceptualization, funding acquisition. PH: supervision, funding acquisition. BAC: funding acquisition, supervision and writing. AEC: funding acquisition, supervision and writing. SS: supervision and writing. JCC: conceptualization, experiments, data processing, data analysis, software development, software testing, writing and supervision.

## Competing interests

The authors declare the following competing interests: SS and AEC serve as scientific advisors for companies that use image-based profiling and Cell Painting (AEC: Recursion, SyzOnc, Quiver Bioscience; SS: Waypoint Bio, Dewpoint Therapeutics, Deepcell) and receive honoraria for occasional talks at pharmaceutical and biotechnology companies. PH is the founder and a shareholder of Single-Cell Technologies Ltd. JCC is a co-founder and shareholder of Quantiscope Ltd. All other authors declare no competing interests.
