## [Peer Review File · Nature Communications]

REVIEWER COMMENTS

Reviewer #1 (Remarks to the Author):

Paper Summary: The authors present a framework to learn cell image features from fluorescence images and test the robustness of the approach by validating against technical confounders and phenotypic variation in three datasets. The learned features are evaluated in the biological matching task by simulating queries to retrieve matching phenotypes. The results show outstanding profiling performance for the downstream analysis tasks when compared with CellProfiler, Imagenet-based features and single-dataset based features.

Paper Strengths:

- The paper is well written.
- The experiments are comprehensive and each of the stages of the approach is properly referenced or explained.
- The problem is introduced clearly and well motivated.
- Figures are illustrative of the concepts involved.
- Source code and datasets for the experiments are provided, ensuring the reproducibility of the approach.

Paper weaknesses:

- A better comparison with other weakly supervised techniques is lacking.
- Some figures lack a detailed explanation and might be better presented when splitted in two or more figures.

Detailed and constructive comments for the authors

- The authors state that classical features and pre-trained networks may not have sufficient expressive power to realise that potential. Can you reference a study where networks pre-trained with natural images are shown to be insufficient to provide an acceptable sensitivity for downstream analysis tasks?

- In figure 5.B the input embedding dimension for the UMAP algorithm is not clearly stated (I suppose is 672?) . Also, is this plot robust to the initial dimension and layer of the features extracted (i.e. would you obtain a similar plot with the features from conv5c or conv6b)?

- Can you elaborate on why the confounder factors are reduced to the batch effects? Are there those the only confounding factors involved in Figure 1.B?

- Did you try other weakly supervised learning techniques or loss functions, if not why? I would have expected to see the comparison with more modern WSL techniques such as multiple instance learning or attention mechanisms.

- Did you compare the sphering batch-correction with other mechanisms to overcome these confounding factors, for example using domain adversarial learning to restrict the influence of those features in the overall representation? Or maybe risk invariant minimisation games? Given that the batch effect is so important to correct, I would have expected to see the comparison with other techniques, particularly non-linear based ones.

- Can you please provide more qualitative results on the effect of the confounding factors in the downstream task? Maybe performing saliency analysis (e.g. Grad-CAM) of the corrected features and the uncorrected ones?

- Are all the features compared in figure 5.A using the batch-correction? If it does, please specify it on the text, if not, please provide the respective plot where all features use the sphering batch-correction.

Reviewer #2 (Remarks to the Author):

Moshkow et al. describe a new machine learning process that can improve predictions from phenotypic profiling data. Let me start by acknowledging that I am not experienced enough in ML to give feedback about the appropriateness of the approach.

However, I liked how the authors described the rationale for many choices, as opposed to simply stating them. They are also transparent about data sources used and make code publicly available. They are aware of different levels of biological variability and tried to include as many confounding factors as possible.

If I understood it correctly, they used from the multiple studies a subset of images only. For chemical treatments with multiple test concentrations, they selected the highest concentration. In my opinion, this has several disadvantages: (1) It biases the models to recognize large phenotypic effects, such as cell death, or other dramatic cell events, over more subtle, possibly very specific phenotypes, that – in my opinion – could contain more valuable biological information. This would reduce the “sensitivity” of the approach. (2) For drug-like chemicals, the phenotype at the highest tested dose might not be associated with their annotated mode-of-action, and rather with “side effects” such as cytotoxicity. For example, estrogens activate estrogen receptors at picomolar concentrations, while at micromolar concentrations they affect microtubules, hence a different phenotype would be expected. This could then decrease the overall prediction, because such a chemical might not be correctly classified.

I would be interested to see in the future how well the presented method works for subtle phenotypes.

Reviewer #3 (Remarks to the Author):

The authors describe a CNN learned on Cell Painting readouts and show they are able to predict/cluster related cell phenotypes and modes of action

It's generally a good paper and helps deal with cell morphology readouts in practice

This referee would only like to make sure a few items about the paper have been done properly:

The authors in the paper have used two validation strategies, leave-plate-out and leave-cell-out:

-> Leave-plate-out: fails to generalise to the validation set

-> Leave-cell-out: a CNN can accurately learn to classify single cells from all plates and treatments in the experiment for training, and leaves a random fraction out for validation

Maybe I missed this part in methods, but if leave-cell-out uses cells from all plates in the dataset for training and leaves a random fraction out for validation it seems the authors used all replicates, hence the same treatment exists at both the replicate and validation levels, resulting in an information leak. Please check that this is not the case, thank you. The obvious solution is to split at compound perturbation level, before splitting cell-out and training a CNN model.

Do the authors perform any interpretation of CNN models to check which part of the cell images the models actually get the most information out of? eg. <https://arxiv.org/abs/2205.10838> provides a way to identify what parts of an image contribute most to the output of a classifier deep network, which can show that models are not just predicting apoptosis (or dark images from well-lit images only, or similar spurious results).

That strong phenotypic effects improve performance is not too surprising - this referee suspects this is why also e.g. gene expression-driven drug repurposing works, despite noisy data; strong signals are retained and true signal, so that makes sense

Figure 5 shows clustering of related MoAs - is this true across scaffolds; maybe bring into context of chemistry as well?

So generally a good paper - please just check the above and then this should be good to go, cheers!

Reviewer #4 (Remarks to the Author):

The paper proposes a deep learning approach to the extraction of biological phenotypes from microscopy images. The workflow builds on top of existing cytometry workflows such as popularized by CellProfiler. Subsequent to conventional segmentation, the main advance/novelty here is to replace the hand-crafted cell features (think eccentricity, Haralick, Zernike moments, ...) with outputs from a convolutional neural network.

This general idea is reasonable and plausible, and the results are promising. The presentation of the work however seems preliminary:

- The paper is in many parts difficult to read because of vagueness and lack of detail where specifics are needed.
- Some methodical choices seem ad hoc (several examples below) and are not well motivated neither from a theoretical nor from a bottomline results point of view.
- The claimed performance improvements over the shallow learning state-of-the-art seem marginal, especially given the fact that they could be optimistic if they were repeatedly measured in parallel to study design and method development (i.e., a kind of “over-fitting” of architecture and implementation details, dataset choices, quality thresholds, etc.) .

Performance comparisons are only made with CellProfiler, a method by some of the same authors. I am not an expert in this particular field, but e.g. a Google search for "deep learning" "cell morphology" yields numerous results and I would expect an effort here at more comprehensive, more systematic benchmarking.

I have only had time to browse <https://github.com/cytomining/DeepProfiler>, not to try install and run. Based on that, data and code availability seem exemplary.

Specific comments:

1. Abstract: it is stated that the three test data sets are public, whereas no such statement is made for the five training datasets. This should be clarified. More generally, a table like Fig. 4B should be given for all datasets used, including specifics like number of images, number plates, dyes used, number and type of perturbations, image resolution and sizes, DOI of the relevant paper, database accession number or URL of the dataset, etc.

2. At four places it is claimed that the methods in this paper were “optimized” and “optimal”. I do not believe this to be true: (1) some method choices seem rather ad hoc and pragmatic, which is fine, but this means no optimisation was performed, (2) no explicit optimality criterion is stated, and (3) it would imply that no further work in this field is necessary, because we already reached the optimum. Perhaps less boastful language like “improved” would be more appropriate.

3. It would be helpful for reviewers if pages and lines could be numbered.

4. The description of “the causal graph” in the first paragraphs of both the Results and Methods sections and in Fig.1B is confusing and as far as I managed to understand, incoherent. A graph is formally defined as a set of nodes (a.k.a. vertices) and a set of edges (see Wikipedia or a maths book). At first, I thought a graph with 4 nodes and 5 directed edges was discussed here, but then the language switches and talks about individual nodes in the plural (“observed images (O)”, “treatments (T)”). I think it would be necessary to be far more explicit about the specific mathematical structures (matrices? functions, from which space into which?) hiding behind the nodes and edges in Fig.1B, at least in the Methods.

- How many confounders are there, and what are they? Plate numbers? Well coordinates within plate? What else?
- How many interventions are there?
- How many outcomes are there, and in which space do they live?
- How many images?

5. Fig. 1C-E (and perhaps some others) seem trivial and I am not sure what value they add.

6. The result presented in Fig.2 is interesting —I think it deserves a much broader discussion and is dealt with too superficially currently. I.e., what sources of noise do “plates” stand for? Is really each plate different, or are there groups of plates that are more similar and then others that are outliers? Are all the plates of equally good data quality, or did some maybe simply fail and should better be dropped for QC reasons rather than “batch-corrected”?

7. The mean average precision values reported in Fig.1D seem to be very small, they take values of ca. 0.17, 0.09, 0.06 on a scale between 0 and 1. Are we here not just scraping the bottom? I.e., if both methods perform equally bad, it does not have to mean that the batch correction has improved anything, it may just as well as inject so much noise that everything is equally bad afterwards. (Leveling out differences is not the same as improving a situation.)

8. “To measure the phenotypic strength of treatments we calculate the Euclidean distance between control and treatment profiles in the CellProfiler feature space” —why Euclidean distance here, while cosine distance is used elsewhere? Maybe it does not matter much, but such choices do leave room for the “method overfitting” I mentioned above, and they may also be inconsistent with “optimality” (Point 2).

9. I was bemused by the statement “However, since we do not observe the control and treated condition in the same cells, this remains only an approximation of the ATE, even if the cells are isogenic clones of each other”. Is somebody reinventing the history of science... We can *never* compare effects on the same cells (or on the same anything), as already Heraclitus realized 2500 years ago (“No man ever steps in the same river twice. For it’s not the same river and he’s not the same man.”). The whole point of the scientific method, of replication and reproducibility, of statistics and sampling theory, developed over the last 500 years or so, is to deal with this challenge, and you do not need to reinvent it here for a special case. Instead, please more explicitly state the generality and replicability level of your findings.

10. The first paragraph of Section “A training set with highly diverse experimental conditions” reads like a complex Discussion section in the middle of what should be Results, and signals hasty writing / poor editing. I suggest shortening (a Results section should work without such lengthy flashbacks).

11. “We created a combined training resource by collecting strong treatments from five different dataset sources, including the three benchmarks evaluated in this work plus two additional publicly available Cell Painting datasets (Figure 4).” I could not really follow what was done here—now the benchmark data from the preceding results is turned into training data? Fig. 4B only lists 3 datasets (with far too little detail, see Point 1) while the text mentions 5. How are the “strong” treatments selected? Top 20% in each case? How do you know that using the same 20%-threshold in each dataset is “optimal”, or even only appropriate?

12. “According to the MAP metric, a WSL model trained on the highly diverse combined set improves performance 7%, 8% and 25% relative to CellProfiler features on BBBC037, BBBC022 and BBBC036 respectively (difference of cyan points vs pink points in the x axis of Figure 2B).” I assume Fig.3D is meant here (or is it 5A?). I could not really follow the 8% claim for BBBC022 (the difference looks smaller in the graph), but in any case these improvements seem pretty minor, at an overall low level, i.e. with both reference and new method performing rather mediocly.

13. “The sphering transform, while effective, is still far from perfect, and further research is needed to better disentangle confounding from phenotypic variation, potentially using nonlinear transformations.” I fully agree, and this one of the reasons for my calling this work “preliminary” in the overall assesment above.

14. The Methods section is in many places too conceptual and philosophical. One wants concrete implementation details and specifics here.

15. Methods 3.1: Is the compression really an important, central part of the method? It seems pretty convoluted, for an overall gain of a factor of 6, which is well within the order of magnitude of typical variations in dataset size, or the variability of available hardware resources in different labs. To me it seems more like a one-off hack.

16. “Stretch the histogram of intensities of each image by removing pixels that are too dark or too bright (below 0.05 and above 99.95 percentiles).” This is self-contradictory. A fixed number of pixels is removed (0.1%), whether these, and only these, are “too dark” or “too bright” is dataset-dependent, and a more data-adaptive threshold (if any) should be used.

17. Methods 3.3: “Random crop and resize”—why is this appropriate? It seems to imply that variations in cell size are not biologically informative. While in fact the phenotypes of many cell cycle or metabolic genes comprise such variation and are biologically informative.

18. “Random horizontal flips and 90-degree rotations.” Why not general rotations (by any angle)? Just singling out these specific symmetry operators seems grossly incomplete.

19. “Random brightness and contrast adjustments, each channel is augmented and renormalized separately.”—again this seems to imply that brightness is not biologically informative, whereas experience from many cellular screens is that important phenotypes are affecting it.

20. (Personally I also find such data augmentation ugly compared to using equivariance, but I understand “it’s what people do”.)

21. Methods 3.5: “As ... images are five-channel and ImageNet pre-trained models expect three-channel (RGB) images, we follow the well established practice of computing a pseudo RGB image for each grayscale fluorescent channel by replicating it three times before passing it through the model (thus each cell requires five inference passes).” — This seems very ugly. Can you provide citations for the “well established practice”? Doesn’t this waste the opportunity of using correlations between the channels? Instead, e.g., why not use all possible 10 subsets of size 3 of the set of 5 channels (5 choose 3 is 10)? That way, the network would at least have the chance to directly access correlations between channels.

22. Methods 4.2: “The feature extraction steps are described in the CellProfiler pipelines published together with the corresponding original datasets. —Sorry for the harshness but this sounds lazy. Please describe what was done here in a self-contained form, perhaps helped by a table. Where appropriate, provide specific citations.

23. Methods 4.3: “Features of single-cells are first aggregated using the median operator at field-of-view (image) level. Next, fields-of-view features are aggregated using the mean to create a well-level profile. Finally, treatment-level profiles are obtained with the average across replicate wells.” — why this combination of medians and means? Why not always median or always mean? This is one of several places where the chosen approach seems to contain arbitrary choices (i.e. was not optimized). Maybe it does not matter, but the way it’s implemented seems a bit chaotic.

24. Methods 4.4: “Batch correction with the sphering transform”—is this more than just a clumsy way to implement a Mahalanobis distance (perhaps with some regularization)? In any case this seems like textbook stuff and should be appropriately cited.

We would like to thank the Editor and Reviewers for their careful and thoughtful consideration of our manuscript. We appreciate their constructive feedback and suggestions, which have helped us to improve the quality of our work. We have addressed each of the reviewers' comments in detail in our responses in blue below.

We are confident that these changes have made our manuscript stronger and more informative. We are grateful to the Editor and Reviewers for their time and expertise, and for being invested in making our manuscript more clear and understandable.

Reviewer #1 (Remarks to the Author):

Paper Summary: The authors present a framework to learn cell image features from fluorescence images and test the robustness of the approach by validating against technical confounders and phenotypic variation in three datasets. The learned features are evaluated in the biological matching task by simulating queries to retrieve matching phenotypes. The results show outstanding profiling performance for the downstream analysis tasks when compared with CellProfiler, Imagenet-based features and single-dataset based features.

Paper Strengths:

- The paper is well written.
- The experiments are comprehensive and each of the stages of the approach is properly referenced or explained.
- The problem is introduced clearly and well motivated.
- Figures are illustrative of the concepts involved.
- Source code and datasets for the experiments are provided, ensuring the reproducibility of the approach.

We thank the reviewer for their comments and for the suggestions to improve the manuscript. We are pleased that the reviewer found that the paper is well-written, properly explained, and well motivated.

Paper weaknesses:

- A better comparison with other weakly supervised techniques is lacking.
- Some figures lack a detailed explanation and might be better presented when splitted in two or more figures.

We appreciate the reviewer's feedback, which we have incorporated in a new version of the manuscript to improve the quality and clarity of our work.

Detailed and constructive comments for the authors

1.1 The authors state that classical features and pre-trained networks may not have sufficient expressive power to realise that potential. Can you reference a study where networks pre-trained with natural images are shown to be insufficient to provide an acceptable sensitivity for downstream analysis tasks?

We regret this confusion. The statement refers to pre-trained networks using natural images. Several proof-of-concept studies found that networks pre-trained with natural images provide acceptable performance, comparable to hand-crafted features (Caicedo et al. 2022; Schiff et al. 2022; Bao et al. 2022; Michael Ando, McLean, and Berndl 2017; Pawlowski et al. 2016). Intuitively, representation learning has the potential to uncover unique features in specialized domains through direct training or fine-tuning with data from those domains. Therefore, researchers have asked whether training can improve downstream performance, which is also the question we address in our work. Studies concurrent to ours have also compared pre-trained models to trained or fine-tuned models, and have found that training and fine-tuning helps realize better performance in applications such as cancer tissue classification (Li et al. 2022), and cellular images (Kensert, Harrison, and Spjuth 2019), including high-throughput Cell Painting images (Wong et al. 2022; Kim et al. 2023). We have added these references to the introduction of the main manuscript as follows:

[...] feature representations need to be sensitive to subtle changes in morphology. Researchers have found that training or fine-tuning networks with high-throughput images can improve downstream performance compared to models trained for natural images (Li et al. 2022; Kensert, Harrison, and Spjuth 2019; Kim et al. 2023; Wong et al. 2023). This indicates that representation learning can identify domain-specific features from cellular images in a data-driven way (Caicedo et al. 2018; Lu et al. 2019; Hofmarcher et al. 2019; Yang et al. 2019; Cuccarese et al. 2020), which also brings unique challenges to prevent confounding factors (Mao et al. 2021; Schölkopf et al. 2021).

1.2 In figure 5.B the input embedding dimension for the UMAP algorithm is not clearly stated (I suppose is 672?). Also, is this plot robust to the initial dimension and layer of the features extracted (i.e. would you obtain a similar plot with the features from conv5c or conv6b)?

The reviewer is correct. The input embedding dimension is 672, and it is now explicitly stated in the figure caption. The UMAP plot generally reveals the structure of the dataset according to the underlying feature embeddings. We have observed that the plots are consistent across several choices of layers and feature extractors, and show similar clusters even with classical features. The differences are difficult to assess qualitatively, which may result in subjective interpretations. For this reason, we instead used quantitative performance metrics to compare features from different layers (Supplementary Figure 8B), which are a proxy indication of how much relevant biological variation could be revealed in a UMAP if it is created with such features. We updated the manuscript as follows:

From a qualitative perspective, the UMAP plot [...] show[s] many treatments grouped together according to their mechanism of action (MoA). The clustering is generally consistent across several choices of layers used for feature extraction, resulting in similar groups from a qualitative point of view. Some of these clusters are also prominent in the chemical feature space for compound perturbations, suggesting complementarity for correctly connecting MoAs (Supplementary Figure 4). The quantitative performance metrics offer a more accurate picture of the biological relevance of these clusters (Supplementary Figure 8B).

1.3 Can you elaborate on why the confounder factors are reduced to the batch effects? Are there those the only confounding factors involved in Figure 1.B?

The reviewer is right that batch effects are not the only confounding factors involved in the analysis of high-throughput images. There are also well-position effects, and plate effects, among others, which are hierarchically organized sources of variation that typically confound the analysis. In the literature, all these sources of variation are usually grouped and called *batch effects*, because addressing them separately is still an open research problem. In our work, we do not make explicit distinction of what is the source of the confounding factors, and for that reason, we model it as an unobserved latent variable.

To avoid any confusion, we have added this clarification to the main text and in the methods section, and changed the reference to batch effects in Figure 1.B as an example only (e.g. batch effects).

Main text:

Note that confounders can include a wide range of technical / nuisance variation, which we group together and refer to as batch effects to be consistent with the related literature.

Methods:

Note that confounders involve a wide range of technical and unwanted variation, not limited to batch effects only. There are also well-position effects, and plate effects, among others, which are hierarchically organized sources of variation that typically confound the analysis. In the literature, all these sources of variation are usually grouped and called batch effects, because addressing them separately is still an open research problem. In our work, we do not make explicit distinction of what is the source of the confounding factors, and for that reason, we model it as an unobserved latent variable.

1.4 Did you try other weakly supervised learning techniques or loss functions, if not why? I would have expected to see the comparison with more modern WSL techniques such as multiple instance learning or attention mechanisms.

We thank the reviewer for suggesting this comparison. We have now implemented two additional weakly supervised learning techniques in addition to the method used in our study, and evaluated them under the same experimental framework. The two techniques are: online label smoothing (Zhang et al. 2021) and multiple-instance learning with attention (Ilse, Tomczak, and Welling 2018). While the formulations are different, we found that they do not improve feature representations. The results have been added in the Supplementary Figure 1, and referenced in the manuscript.

We hypothesize that the main reason for this lack of improved performance when using more complex techniques is that they are designed to improve the auxiliary classification task rather than the feature representations. In a validation experiment comparing leaving cells out and leaving plates out, we observed that improving the performance of the auxiliary classification task does not result in better downstream analysis performance (Supplementary Figure 1). We conclude that to obtain better representations, the specific classification algorithm used for auxiliary training does not seem to play an important role. We added this insight in the main text as follows:

The same effect is observed with alternative WSL approaches. Our experiments are based on an EfficientNet model trained with a classification loss and weak labels. We explored the use of other loss functions that have the potential to improve performance in the presence of weak labels: Online Label Smoothing (Zhang et al. 2021) and Multiple Instance Learning with Attention (Ilse, Tomczak, and Welling 2018) (Methods). We did not observe improved results when using these alternative WSL formulations (Supplementary Figure 1), primarily because these methods are designed to improve the performance of a classifier with weak or noisy labels, instead of learning disentangled representations. We use the classifier as a pretext training component, but the main problem to be solved is learning representations that are invariant to confounding factors.

This also confirms that one of the main challenges is still to address the confounding factors, which was also pointed out by the reviewer in the next question.

1.5 Did you compare the sphering batch-correction with other mechanisms to overcome these confounding factors, for example using domain adversarial learning to restrict the influence of those features in the overall representation? Or maybe risk invariant minimisation games? Given that the batch effect is so important to correct, I would have expected to see the comparison with other techniques, particularly non-linear based ones.

We thank the reviewer for suggesting this evaluation. We have now implemented two additional batch-correction mechanisms to evaluate the impact in downstream performance. In particular, we use the gradient-reversal layer (GRL) (Ganin and Lempitsky 2015), and Harmony (Korsunsky et al. 2019). GRL is a domain adaptation, adversarial learning algorithm applied during training, while Harmony is a batch-integration algorithm applied after training. We applied both with and without sphering, which is our baseline correction method, and the results indicate

that there is no major performance difference. We report the results in the manuscript as follows:

We evaluated two alternative batch correction methods that use nonlinear transformations of features to remove unwanted variation: Harmony (Korsunsky et al. 2019) and Gradient Reversal Layer (Ganin and Lempitsky 2015) (GRL). The results indicate that these nonlinear batch correction methods do not improve the performance in the downstream analysis task, and in fact are unable to match the performance of sphering (Supplementary Figure 5). The main reason is that Harmony and GRL require prior knowledge about what the unwanted source of variation is. In high-throughput imaging, the technical sources of variation are complex and hierarchically organized, and unfortunately, these methods make the assumption that there is a single source of confounding variation organized categorically (batch labels). Sphering is more effective because it makes no such assumption, and instead uses control cells to model technical variation in a non-parametric way.

Finally, we thank the reviewer for suggesting risk invariant minimization games (Ahuja et al. 2020). We have added the reference with other potential techniques that can be used for batch-correction (Lin and Lu 2022) to the manuscript and will leave the evaluation for future work:

This is an active research area with novel solutions being explored, including multi-layer normalization strategies (Lin and Lu 2022), and the use of generative models (Pernice et al. 2023). Other potential solutions may include invariant risk minimization games (Ahuja et al. 2020) or similar formulations.

1.6 Can you please provide more qualitative results on the effect of the confounding factors in the downstream task? Maybe performing saliency analysis (e.g. Grad-CAM) of the corrected features and the uncorrected ones?

Unfortunately, it is not possible to perform saliency analysis with Grad-CAM for corrected features (as used in the UMAP visualizations and the downstream analysis) first, because these have been aggregated and no longer correspond to single cells, and second, because these have been transformed with sphering, which is not a computation layer of the neural network. Therefore, we designed a Grad-CAM experiment to visualize salient features in DMSO cells across wells and plates, as well as for different treatments (Supplementary Figure 6). We found that the saliency maps are very homogeneous in our experiments, and report the findings in the main text as follows:

To further investigate what are the features that characterize treatments or batch effects, we ran a Grad-CAM analysis on a sample of cell images in the BBBC022 dataset (Supplementary Figure 6). We did not observe any major indication of features that can be localized in the 2D image plane at the single-cell level that could reveal small

differences in technical variation (Supplementary Figure 6). Batch effects appear more prominently after aggregating single-cell features into population level profiles (Supplementary Figure 7). This suggests that technical variation (such as wells, plates, and batches) may be accumulated during the aggregation steps while single-cell heterogeneity may be smoothed out. Thus, correcting batch effects and separating biological from technical variation may be more efficient when considering information at various resolutions jointly (single-cells, image-level, well-level, treatment-level).

And added more details in the Supplementary Material:

A saliency analysis of features was performed using the Grad-CAM algorithm. The goal of this experiment was to investigate if there are prominent features that characterize treatments or confounding factors. For this experiment we focused on an EfficientNet model trained with weakly supervised learning only on the BBBC022 dataset (leaving cells out), assuming that it is sensitive to technical variation specific to that dataset. We selected a subset of single cells from the validation set where the network prediction was correct, and then ran Grad-CAM to generate saliency maps that explain the classification. The layer used for analysis is block6a, which is where features are extracted from for downstream biological tasks. We started with cells from the DMSO control that come from different wells and plates and asked the question: does the model focus on the same structures to classify controls or are these specific to the plate or well? We also looked at treated cells from different plates for comparison (Supplementary Figure 6).

The saliency maps in all cases tend to concentrate attention in the central cell of the image, and sometimes attends a secondary cell or part of the context. There is no major or obvious difference between the images, and the saliency maps do not reveal a significant structure of interest. Unfortunately, Grad-CAM makes the assumption that features are localized in the 2D space, but the differences may be happening selectively in individual channels. Given how homogeneous the saliency maps are across plates and treatments, we conclude that it is difficult to interpret features using this approach. We recommend using quantitative evaluations to objectively assess the performance of models, and to prevent drawing conclusions subjectively from a few examples.

1.7 Are all the features compared in figure 5.A using the batch-correction? If it does, please specify it on the text, if not, please provide the respective plot where all features use the sphering batch-correction.

We thank the reviewer for pointing this out. We confirm that the results presented in Figure 5A were obtained with batch-corrected features. We have fixed the caption and it is now explicitly stated.

Reviewer #2 (Remarks to the Author):

Moshkov et al. describe a new machine learning process that can improve predictions from phenotypic profiling data. Let me start by acknowledging that I am not experienced enough in ML to give feedback about the appropriateness of the approach.

However, I liked how the authors described the rationale for many choices, as opposed to simply stating them. They are also transparent about data sources used and make code publicly available. They are aware of different levels of biological variability and tried to include as many confounding factors as possible.

We thank the reviewer for the comments and feedback. We appreciate that they find our explanations descriptive and helpful.

If I understood it correctly, they used from the multiple studies a subset of images only. For chemical treatments with multiple test concentrations, they selected the highest concentration. In my opinion, this has several disadvantages: (1) It biases the models to recognize large phenotypic effects, such as cell death, or other dramatic cell events, over more subtle, possibly very specific phenotypes, that – in my opinion – could contain more valuable biological information. This would reduce the “sensitivity” of the approach.

We thank the reviewer for pointing this out. There are two main distinctions that need to be clarified: the concentration of compounds and the strength of phenotypes - although causally related, are different. We added the following explanation in the Methods section:

There is an important distinction between compound concentration and phenotypic strength. Although they are causally related, they are different concepts. First, the concentration of compounds was predetermined as part of the experimental design. Two of our three evaluation datasets (BBBC022 and BBBC036) are compound screens, and the third dataset is a gene overexpression experiment at a single “dosage” (BBBC037). The compound screens prioritized diversity of compounds with a fixed dose instead of multiple concentrations for titration studies. Therefore, for most of the compounds in BBBC022 and BBBC036 only a single concentration was available, usually a high one. Second, the strength of phenotypes is the effect that perturbations produce on cells, regardless of their type or concentration. A high dose does not mean that the phenotypic effect is strong, because some compounds may not induce a phenotypic effect that can be detected with imaging, even at high concentrations. Therefore, it is important to separate the cause (compound dose) from the effect (phenotypic change) to correctly interpret the results.

The main difficulty in identifying subtle phenotypic differences is that there are other subtle differences that are not biologically relevant. To facilitate that the models learn relevant biological variation, we selected compounds with strong phenotypic effects for training. As noted

by the reviewer, this indeed may bias models to recognize only large phenotypic effects, reducing the sensitivity of the approach. We discuss this in the Results section as follows:

Note that strong phenotypic effect (large Average Treatment Effect - ATE) is not the same as having high compound concentrations (Methods). Detecting treatments with subtle phenotypic effects is challenging in any platform, including imaging and gene expression (Subramanian et al. 2017a; Way et al. 2022a). Selecting treatments with strong phenotypic effects serves as a strategy to trade-off the sensitivity to subtle biological variation and the introduction of confounding variation. When confounding variation is stronger than subtle phenotypes, WSL models will preferentially rely on the most prominent signal (confounding). Removing treatments with weak phenotypic effect can break this dependency, resulting in models with better performance. The threshold that separates strong vs weak phenotypes was selected in our study based on the percentiles of the ATE distribution, and it can be used as a mechanism to decide how much biological or confounding variation the model observes during training.

(2) For drug-like chemicals, the phenotype at the highest tested dose might not be associated with their annotated mode-of-action, and rather with “side effects” such as cytotoxicity. For example, estrogens activate estrogen receptors at picomolar concentrations, while at micromolar concentrations they affect microtubules, hence a different phenotype would be expected. This could then decrease the overall prediction, because such a chemical might not be correctly classified.

This is a great point. We used mode-of-action (MoA) annotations publicly available in compound libraries to estimate performance in a quantitative fashion. However, the ground truth annotations may not be 100% accurate all the time for several reasons, including the dose response pointed out by the reviewer. This is also another reason why we emphasize weakly-supervised learning rather than fully supervised learning: these annotations are a guide to assess performance, and they should not be used for directly training machine learning models. We clarify this in the Methods section as follows:

As ground truth annotations for evaluating downstream biological tasks, we used mechanism of action (MoA) labels publicly available in compound libraries. These annotations may not be 100% accurate for several reasons; for example, the phenotype of compounds may vary depending on the tested dose resulting in a different MoAs for the same compound. The MoA annotations have not been manually and individually confirmed for each compound in the study; they only represent a potentially expected phenotype according to what we know in the literature about these compounds. Factors such as the dosage, the sensitivity of imaging, and confounders, among others may determine whether the MoA association is correct or not. The quantitative evaluation presented in this study is an attempt to measure how likely two related phenotypes are given the observations and the available annotations.

There is so much more to be discovered about phenotypic response using images, and our study proposes strategies to improve quantitative methods to facilitate future discoveries.

I would be interested to see in the future how well the presented method works for subtle phenotypes.

We thank the reviewer for this suggestion. We agree that characterizing subtle phenotypes is an important endeavor, and we have added this to our discussion of future work as follows:

Quantifying subtle phenotypic effects remains an ongoing challenge. We successfully trained a model that improved performance after using treatments with strong cellular response, which helped reduce the impact of technical variation on downstream tasks. However, this still does not recover the phenotypic signal of all treatments, and the list of treatments with weak effects remains long. The question whether such effects are overpowered by unwanted variation or if they can even be detected by imaging is still open. Our work presents a strategy to learn representations under real world conditions, and a benchmark to continue study subtle phenotypic effects under noisy conditions.

Reviewer #3 (Remarks to the Author):

The authors describe a CNN learned on Cell Painting readouts and show they are able to predict/cluster related cell phenotypes and modes of action
It's generally a good paper and helps deal with cell morphology readouts in practice

This referee would only like to make sure a few items about the paper have been done properly:

3.1 The authors in the paper have used two validation strategies, leave-plate-out and leave-cell-out:

-> Leave-plate-out: fails to generalise to the validation set

-> Leave-cell-out: a CNN can accurately learn to classify single cells from all plates and treatments in the experiment for training, and leaves a random fraction out for validation

Maybe I missed this part in methods, but if leave-cell-out uses cells from all plates in the dataset for training and leaves a random fraction out for validation it seems the authors used all replicates, hence the same treatment exists at both the replicate and validation levels, resulting in an information leak. Please check that this is not the case, thank you. The obvious solution is to split at compound perturbation level, before splitting cell-out and training a CNN model.

We are happy to have the opportunity to explain the lack of information leak in this context. The reviewer is correct that we used all replicates in the leave-cells-out experiment, as described in the main text:

The leave-cells-out validation scheme uses single cells from all plates and treatments in the experiment for training, and leaves a random fraction out for validation. By doing so, trained CNNs have the opportunity to observe the whole distribution of phenotypic features (all treatments) as well as the whole distribution of confounding factors (all batches or plates). In contrast, the leave-plates-out validation scheme separates different technical replicates (plates) for training and validation, resulting in a model that still observes the whole distribution of treatments, but only partially sees the distribution of confounding factors.

We make the following clarification at the end of the Section “Weakly supervised learning captures confounders and phenotypic outcomes of treatments”:

Note that using all replicates in the leave-cells-out validation experiment does not result in information leaks with respect to the downstream biological task. Compounds are always known ahead of time, which is the information the models use for training. What we assume to be unknown is the mode of action (MoA), which is information always left out for downstream evaluation and never seen by models during training. This approach measures how well the MoA information emerges from learned representations instead of directly training for MoA classification. Similar evaluation setups have been also used in previous work (Caicedo et al. 2018; Cross-Zamirski et al. 2022).

In practice, if MoA inference was perfect, the method would be useful for biological discovery even if we need to train models with all compounds. This is not the case yet, and in our study we aim to understand why.

Leaving compounds out of the training set, as the reviewer suggests, would be useful to evaluate how well models infer MoA on new compounds not used during training, which is a more challenging generalization task. Inferring MoAs on compounds not seen during training suffers from potential confounders because new compounds come from new plates. Artificially splitting compounds that come from the same batch or plate for training and validation results in shared technical variation (plate or position effects), thus a confounded experiment.

3.2 Do the authors perform any interpretation of CNN models to check which part of the cell images the models actually get the most information out of? eg. <https://arxiv.org/abs/2205.10838> provides a way to identify what parts of an image contribute most to the output of a classifier deep network, which can show that models are not just predicting apoptosis (or dark images from well-lit images only, or similar spurious results).

We thank the reviewer for this suggestion. We used GradCAM, as suggested, to analyze a model trained on compound perturbations by visualizing image regions that contribute the most to the output classifier. We found that the saliency maps are very homogeneous in our experiments, and cannot be generally used to confirm or reject spurious correlations in the

predictions. Part of the reason is that saliency maps are a qualitative analysis dependent on specific examples. We report the findings in the main text as follows:

To further investigate what are the features that characterize treatments or batch effects, we ran a Grad-CAM analysis on a sample of cell images in the BBBC022 dataset. We did not observe any major indication of features that can be localized in the 2D image plane at the single-cell level that could reveal small differences in technical variation (Supplementary Figure 6).

We added more details in the Supplementary Material:

A saliency analysis of features was performed using the Grad-CAM algorithm. The goal of this experiment was to investigate if there are prominent features that characterize treatments or confounding factors. For this experiment we focused on an EfficientNet model trained with weakly supervised learning only on the BBBC022 dataset (leaving cells out), assuming that it is sensitive to technical variation specific to that dataset. We selected a subset of single cells from the validation set where the network prediction was correct, and then ran Grad-CAM to generate saliency maps that explain the classification. The layer used for analysis is block6a, which is where features are extracted from for downstream biological tasks. We started with cells from the DMSO control that come from different wells and plates and asked the question: does the model focus on the same structures to classify controls or are these specific to the plate or well? We also looked at treated cells from different plates for comparison (Supplementary Figure 6).

The saliency maps in all cases tend to concentrate attention in the central cell of the image, and sometimes attends a secondary cell or part of the context. There is no major or obvious difference between the images, and the saliency maps do not reveal a significant structure of interest. Unfortunately, Grad-CAM makes the assumption that features are localized in the 2D space, but the differences may be happening selectively in individual channels. Given how homogeneous the saliency maps are across plates and treatments, we conclude that it is difficult to interpret features using this approach. We recommend using quantitative evaluations to objectively assess the performance of models, and to prevent drawing conclusions subjectively from a few examples.

That strong phenotypic effects improve performance is not too surprising - this referee suspects this is why also e.g. gene expression-driven drug repurposing works, despite noisy data; strong signals are retained and true signal, so that makes sense.

We agree with the reviewer that subtle phenotypes are hard to detect in any platform, not just in imaging. We now highlight this in the main text as follows:

Note that strong phenotypic effect (large Average Treatment Effect - ATE) is not the same as having high compound concentrations (Methods). Detecting treatments with subtle phenotypic effects is challenging in any platform, including imaging and gene expression (Subramanian et al. 2017b; Way et al. 2022b). Selecting treatments with strong phenotypic effects serves as a strategy to trade-off the sensitivity to subtle biological variation and the introduction of confounding variation. When confounding variation is stronger than subtle phenotypes, WSL models will preferentially rely on the most prominent signal (confounding). Removing treatments with weak phenotypic effect can break this dependency, resulting in models with better performance. The threshold that separates strong vs weak phenotypes was selected in our study based on the percentiles of the ATE distribution, and it can be used as a mechanism to decide how much biological or confounding variation the model observes during training.

3.3 Figure 5 shows clustering of related MoAs - is this true across scaffolds; maybe bring into context of chemistry as well?

We thank the reviewer for this suggestion! We computed Morgan fingerprints of the chemical structures in the two compound datasets (BBBC022 and BBBC036), and visualized them with a UMAP projection in parallel with the visualization of image-based features using our method (Supplementary Figure 4). We found that MoA clustering is also observable in the chemical structure space, although the organization is different, and seems complementary to phenotypic readouts with imaging. We mention this in the main text as follows:

The clustering is generally consistent across several choices of layers used for feature extraction, resulting in similar groups from a qualitative point of view. Some, but not all, of these clusters are also prominent in the chemical feature space for compound perturbations, suggesting complementarity for correctly connecting MoAs (Supplementary Figure 4). The quantitative performance metrics offer a more accurate picture of the biological relevance of these clusters (Supplementary Figure 8B).

So generally a good paper - please just check the above and then this should be good to go, cheers!

We thank the reviewer for the insightful comments and suggestions!

Reviewer #4 (Remarks to the Author):

The paper proposes a deep learning approach to the extraction of biological phenotypes from microscopy images. The workflow builds on top of existing cytometry workflows such as popularized by CellProfiler. Subsequent to conventional segmentation, the main advance/novelty here is to replace the hand-crafted cell features (think eccentricity, Haralick, Zernike moments, ...) with outputs from a convolutional neural network.

This general idea is reasonable and plausible, and the results are promising. The presentation of the work however seems preliminary:

- The paper is in many parts difficult to read because of vagueness and lack of detail where specifics are needed.
- Some methodical choices seem ad hoc (several examples below) and are not well motivated neither from a theoretical nor from a bottomline results point of view.
- The claimed performance improvements over the shallow learning state-of-the-art seem marginal, especially given the fact that they could be optimistic if they were repeatedly measured in parallel to study design and method development (i.e., a kind of “over-fitting” of architecture and implementation details, dataset choices, quality thresholds, etc.) .

Performance comparisons are only made with CellProfiler, a method by some of the same authors. I am not an expert in this particular field, but e.g. a Google search for "deep learning" "cell morphology" yields numerous results and I would expect an effort here at more comprehensive, more systematic benchmarking.

The reviewer is correct that there are numerous research papers that investigate deep learning for cell morphology - these primarily focus on deep learning for cell segmentation and cell classification. By contrast, our study is focused on representation learning, an approach to capture the broad profiling treatment effects in an unsupervised manner to detect relationships among treatments, where deep learning solutions have started to emerge recently. The vast majority of published Cell Painting studies have been conducted using classical features, thus, we benchmark against this well-established and widely-adopted approach.

Nevertheless, we have now implemented two other weakly supervised learning techniques in addition to the method used in our study, and evaluated them under the same experimental framework. The two techniques are: online label smoothing (Zhang et al. 2021) and multiple-instance learning with attention (Ilse, Tomczak, and Welling 2018). While the methods have different objectives, they do not seem to improve feature representations. The results have been added in the Supplementary Figure 1, and referenced in the manuscript. This suggests that to obtain better representations, the specific classification algorithm used for auxiliary training does not play an important role. We clarified this in the main text as follows:

The same [gap in performance] is observed with alternative WSL approaches. Our experiments are based on an EfficientNet model trained with a classification loss and weak labels. We explored the use of other loss functions that have the potential to improve performance in the presence of weak labels: Online Label Smoothing (Zhang et al. 2021) and Multiple Instance Learning with Attention (Ilse, Tomczak, and Welling 2018) (Methods). We did not observe improved results when using these alternative WSL formulations (Supplementary Figure 1), primarily because these methods are designed to improve the performance of a classifier with weak or noisy labels, instead of learning disentangled representations. We use the classifier as a pretext training component, but the main problem to be solved is learning representations that are invariant to confounding factors.

This also confirms that one of the main challenges is still to address the confounding factors. We did not observe major differences in performance when using more complex training strategies, as reported in these new results. In addition, in early experiments we observed that the architecture of the convolutional network does not impact significantly the results either (data not shown). The model architecture and training strategy are not as important as other components of the methodology for analyzing Cell Painting images. Instead, our finding is that batch correction is the most important aspect to consider for improved performance.

I have only had time to browse <https://github.com/cytomining/DeepProfiler>, not to try install and run. Based on that, data and code availability seem exemplary.

We thank the reviewer for noting that our code and documentation are well organized.

Specific comments:

1. Abstract: it is stated that the three test data sets are public, whereas no such statement is made for the five training datasets. This should be clarified. More generally, a table like Fig. 4B should be given for all datasets used, including specifics like number of images, number plates, dyes used, number and type of perturbations, image resolution and sizes, DOI of the relevant paper, database accession number or URL of the dataset, etc.

We thank the reviewer for this suggestion. The abstract now explicitly mentions the difference:

[...] we constructed a large training dataset with Cell Painting images from five different studies to maximize experimental diversity [... and] We conducted a comprehensive evaluation of our strategy on three publicly available Cell Painting datasets [...]

In addition, we modified Figure 4B to include the two datasets missing plus the information suggested by the reviewer. Now it includes all data sources we used in our study (which indeed are all publicly and freely available). The citations to the publications related to the five datasets and associated URLs are also available in the “Data availability” section.

2. At four places it is claimed that the methods in this paper were “optimized” and “optimal”. I do not believe this to be true: (1) some method choices seem rather ad hoc and pragmatic, which is fine, but this means no optimisation was performed, (2) no explicit optimality criterion is stated, and (3) it would imply that no further work in this field is necessary, because we already reached the optimum. Perhaps less boastful language like “improved” would be more appropriate.

We thank the reviewer for pointing this out. We have removed these statements from the main text and now use “improved” as recommended.

3. It would be helpful for reviewers if pages and lines could be numbered.

We thank the reviewer for the suggestion. This version of the paper has page and line numbers.

4. The description of “the causal graph” in the first paragraphs of both the Results and Methods sections and in Fig.1B is confusing and as far as I managed to understand, incoherent. A graph is formally defined as a set of nodes (a.k.a. vertices) and a set of edges (see Wikipedia or a maths book). At first, I thought a graph with 4 nodes and 5 directed edges was discussed here, but then the language switches and talks about individual nodes in the plural (“observed images (O)”, “treatments (T)”). I think it would be necessary to be far more explicit about the specific mathematical structures (matrices? functions, from which space into which?) hiding behind the nodes and edges in Fig.1B, at least in the Methods.

- How many confounders are there, and what are they? Plate numbers? Well coordinates within the plate? What else?
- How many interventions are there?
- How many outcomes are there, and in which space do they live?
- How many images?

We regret the confusion. We now clarify in the Methods section that each of the nodes in the graph represents a random variable from a probability distribution where instances are sampled, as follows:

Each of the nodes in the graph is a random variable with an associated probability distribution, which is unknown, but can be approximated with data. The following table summarizes the assumptions made in practice for each variable:

Variable	Type	Encoding	Cardinality	Order
Treatments T	Categorical	One-hot vector	$T \in \{0, 1\}^N$	$N \sim \mathcal{O}(10^3)$
Images O	Two dimensional	Continuous pixel intensities in a matrix	$O \in \mathbb{R}^{W \times H}$	$W \times H \sim \mathcal{O}(10^4)$
Phenotypes Y	Latent variable	Continuous multidimensional vector	$Y \in \mathbb{R}^M$	$M \sim \mathcal{O}(10^2)$
Batches C	Categorical	One-hot vector	$C \in \{0, 1\}^K$	$K \sim \mathcal{O}(10^2)$

The exact cardinality of each variable depends on the dataset of focus. A dataset in our study is a particular instance of the graph with a total of P images observed. Therefore, a dataset can be thought of as a matrix with P rows and (N + WxH + M + K) columns, if all the variables were concatenated. In practice, each submatrix is processed by a separate transformation function, notably images are the input to a neural network that predicts the treatments and recovers the phenotype in a hidden feature layer (see Weakly Supervised Learning below for more details).

5. Fig. 1C-E (and perhaps some others) seem trivial and I am not sure what value they add.

We understand that visual representations of technical concepts may seem trivial to expert readers. In our work, we use them to help non-technical readers get a sense of how the methods work, even if they do not follow all the mathematical details. Similarly, we do not expect our technical readers to understand all the biological details. In our manuscript, we aim to keep a balance to inform both audiences because multidisciplinary collaboration is key to continue making progress.

6. The result presented in Fig.2 is interesting—I think it deserves a much broader discussion and is dealt with too superficially currently. I.e., what sources of noise do “plates” stand for? Is really each plate different, or are there groups of plates that are more similar and then others that are outliers? Are all the plates of equally good data quality, or did some maybe simply fail and should better be dropped for QC reasons rather than “batch-corrected”?

We agree with the reviewer that the result is interesting. We confirm that QC was performed as a preparation step before the data was analyzed, and we did not include any outliers (see Methods, “Benchmarks and ground truth annotations”). In addition, we expanded the discussion about this result as follows:

Technical variation manifests in images in subtle ways that cannot be readily distinguished by eye. There is a set of technical factors that includes microscopes, date and time of acquisition, technician, plate-to-plate variation (differences in assay preparation), well-position effects (differences in humidity and temperature), among others. Despite the best efforts to automate and standardize experiment preparation and image acquisition, these factors continue to influence image-based measurements because they reflect microscopic events that cannot be fully controlled. Some of these factors have stronger effects than others, but all of them accumulate in images in unpredictable ways that represent confounding factors. Technical variation may be recorded in images as hidden patterns that are unobservable to the human eye, but computational methods can easily see and incorporate in their metrics. Batch correction methods aim to remove these factors of unwanted variation from image features, and given their unspecified nature, this is still an open research problem.

7. The mean average precision values reported in Fig.1D seem to be very small, they take values of ca. 0.17, 0.09, 0.06 on a scale between 0 and 1. Are we here not just scraping the bottom? I.e., if both methods perform equally bad, it does not have to mean that the batch correction has improved anything, it may just as well as inject so much noise that everything is equally bad afterwards. (Leveling out differences is not the same as improving a situation.)

We agree with the interpretation of precision values reported in Figure 2D suggested by the reviewer. We added this to the main manuscript:

[...] after batch correction, the representations of models trained with leave-cells-out and leave-plates-out yield similar downstream performance in the biological matching task, indicating that both models find similar phenotypic features, but capture different confounding variation. Importantly, batch correction is not improving the situation of either of the two strategies compared in Figure 2D. Instead, batch correction is removing the noise and biases and leveling out the situation of both models. This means that neither of the models learned anything different or more useful than the other despite having drastically different performance on the auxiliary task.

In addition, the low mean-average-precision (MAP) values reflect both the difficulty of the task and the strict evaluation approach that we use. On the one hand, the task is particularly difficult because the phenotypes are very subtle and overpowered by technical variation, and because MoA annotations are used as the best available ground truth even though they are known to be far from perfect for many reasons. Also, the datasets that we consider in this study have thousands of treatments and hundreds of MoAs, which is representative of real-world applications. The chance of matching two compounds correctly is very small (BBBC037= \sim 0.036, BBBC022 = \sim 0.0025, BBBC036= \sim 0.0017), which is better reflected in the folds-of-enrichment metric used in our evaluation. On the other hand, our evaluation protocol is particularly strict by design. We chose to use metrics that are aligned with machine learning practice to reveal the real accuracy of models in a large scale experiment. In addition, we do not train deep learning models to predict the MoA labels used in the evaluation; instead, we train deep learning models to extract features and then we evaluate their biological relevance using the simplest strategy possible: a k-nearest-neighbor classifier. The assumption behind this approach is that all the learning should happen in a weakly supervised fashion, and that the biological information should emerge without further optimization.

Therefore, rather than scraping the bottom, the reported MAP values reflect the increasing difficulty of the tasks in the three datasets, as well as the ample room for improvement that is available for further research. In our experience these numbers are very hard to improve, yet, our proposed methodology resulted in non-trivial improvements as reported in Figure 5.

8. “To measure the phenotypic strength of treatments we calculate the Euclidean distance between control and treatment profiles in the CellProfiler feature space”—why Euclidean distance here, while cosine distance is used elsewhere? Maybe it does not matter much, but such choices do leave room for the “method overfitting” I mentioned above, and they may also be inconsistent with “optimality” (Point 2).

We regret the confusion. We clarify this subtle difference in the Methods section as follows:

To estimate Average Treatment Effect (ATE) we use the Euclidean distance between feature representations obtained with classical features as an independent measure of cellular morphology. The Euclidean distance is a useful estimator of ATE, because it is by definition the expected difference between two outcomes. We approximate this quantity with the Euclidean distance between negative controls and treatments given

that profiles are high-dimensional and we aim to capture the total effect (distance). The cosine similarity was not used in this context because it only measures the directionality of the effect and ignores its magnitude. We consistently use the cosine distance among treatment profiles in all the biological matching tasks involving MoAs (see Methods 4.5 Similarity Matching for more details).

9. I was bemused by the statement “However, since we do not observe the control and treated condition in the same cells, this remains only an approximation of the ATE, even if the cells are isogenic clones of each other”. Is somebody reinventing the history of science... We can *never* compare effects on the same cells (or on the same anything), as already Heraclitus realized 2500 years ago (“No man ever steps in the same river twice. For it’s not the same river and he’s not the same man.”). The whole point of the scientific method, of replication and reproducibility, of statistics and sampling theory, developed over the last 500 years or so, is to deal with this challenge, and you do not need to reinvent it here for a special case. Instead, please more explicitly state the generality and replicability level of your findings.

Well said; we removed that sentence to avoid confusions and misinterpretations.

10. The first paragraph of Section “A training set with highly diverse experimental conditions” reads like a complex Discussion section in the middle of what should be Results, and signals hasty writing / poor editing. I suggest shortening (a Results section should work without such lengthy flashbacks).

We thank the reviewer for pointing this out. We have simplified the paragraph as follows:

Training a model with all the data in an individual dataset does not necessarily improve performance with respect to the baseline (Figure 3D green vs pink points). Changing the distribution of confounders does not change performance in the biological matching task after batch correction (Figure 2D). The only factor that impacted performance was changing the distribution of phenotypic outcomes (Figure 3D). Therefore, we hypothesize that training with data beyond the individual dataset of interest while favoring phenotypic diversity could result in improved performance.

11. “We created a combined training resource by collecting strong treatments from five different dataset sources, including the three benchmarks evaluated in this work plus two additional publicly available Cell Painting datasets (Figure 4).” I could not really follow what was done here—now the benchmark data from the preceding results is turned into training data? Fig. 4B only lists 3 datasets (with far too little detail, see Point 1) while the text mentions 5. How are the “strong” treatments selected? Top 20% in each case? How do you know that using the same 20%-threshold in each dataset is “optimal”, or even only appropriate?

We regret this confusion. We clarify in the manuscript as follows:

To increase the diversity of experimental conditions in the training set, we created a combined training resource by collecting strong treatments from the five dataset sources listed in Figure 4B. We first filtered strong treatments from each source and prioritized treatments shared across sources (Methods and Figure 4A). We selected 348 treatments from BBBC022 (strongest 35%), 354 from BBBC036 (strongest 23%) and 47 treatments from BBBC037 (strongest 23%). We complemented these treatments with the corresponding replicates in the LINCS and LUAD datasets, and added 7 new compounds and 32 new gene overexpression perturbations, resulting in 488 treatments in total (Figure 4A). This combined set also represents two cell lines, two types of negative controls, and examples from more than 200 plates. This results in training data with high experimental diversity with respect to the two latent variables in the causal graph: technical variation (confounders C) and phenotypic variation (outcomes Y).

In addition, we updated Figure 4 to facilitate reading the structure of the resulting dataset. Figure 4 now includes a Sankey Funnel diagram of the resulting training dataset (after combining sources), as well as a table with the details of the sources.

When selecting treatments for this combined set, we fixed a budget to keep the resource computationally manageable in our infrastructure. We aimed for approximately 500 strong treatments (from more than 2,000) that were shared across all five sources. Given the scale of the experiment, we did not repeat this sampling with different thresholds. Our goal was not to optimize the selection of treatments, but rather demonstrate that filtering compounds with weak phenotypic effects out of the training process can help reduce the confounding effects.

Note that the combined dataset contains biological variation that is presumably larger than the technical variation: different cell lines perturbed with strong treatments. Our key observation is that only when the biological variation is more prominent than technical variation, weakly supervised learning can break the link between confounders and outcomes in the causal graph.

12. "According to the MAP metric, a WSL model trained on the highly diverse combined set improves performance 7%, 8% and 25% relative to CellProfiler features on BBBC037, BBBC022 and BBBC036 respectively (difference of cyan points vs pink points in the x axis of Figure 2B)." I assume Fig.3D is meant here (or is it 5A?). I could not really follow the 8% claim for BBBC022 (the difference looks smaller in the graph), but in any case these improvements seem pretty minor, at an overall low level, i.e. with both reference and new method performing rather mediocly.

We regret this confusion. The reviewer is correct that the differences come from Figure 5A. We have fixed the typos and also created a Table in Supplementary Figure 9 with the exact numbers for reference. We also clarify the following:

Many machine learning applications typically aim to replicate human behavior (e.g., classifying cats) with 100% accuracy. However, the purpose of image-based profiling is not to mimic human behavior, but rather uncover new knowledge. The accuracy metrics

reported in our study (Figure 5 and Supplementary Figure 9) are limited by currently available knowledge about these treatments. However, ground truth annotations may be incomplete; uncovering and understanding the mechanisms of these drugs is one of the goals of these studies. The metrics serve the purpose of estimating methodological improvement that can result in future discoveries. Improvements in performance in a drug discovery project could potentially mean new candidate treatments useful for certain diseases.

13. “The sphering transform, while effective, is still far from perfect, and further research is needed to better disentangle confounding from phenotypic variation, potentially using nonlinear transformations.” I fully agree, and this is one of the reasons for my calling this work “preliminary” in the overall assessment above.

We conducted additional experiments with other batch correction algorithms to expand the type of methods evaluated in this work. In particular, we use the gradient-reversal layer (GRL) (Ganin and Lempitsky 2015), and Harmony (Korsunsky et al. 2019). GRL is a domain adaptation, adversarial learning algorithm applied during training, while Harmony is a batch-integration algorithm applied after training. We applied both with and without sphering, which is our baseline correction method, and the results indicate that there is no major performance difference. We report the results in the manuscript as follows:

We evaluated two alternative batch correction methods that use nonlinear transformations of features to remove unwanted variation: Harmony (Korsunsky et al. 2019) and Gradient Reversal Layer (Ganin and Lempitsky 2015) (GRL). The results indicate that these nonlinear batch correction methods do not improve the performance in the downstream analysis task, and in fact are unable to match the performance of sphering (Supplementary Figure 4). The main reason is that Harmony and GRL require prior knowledge about what the unwanted source of variation is. In high-throughput imaging, the technical sources of variation are complex and hierarchically organized, and unfortunately, these methods make the assumption that there is a single source of confounding variation organized categorically (batch labels). Sphering is more effective because it makes no such assumption, and instead uses control cells to model technical variation in a non-parametric way.

We also added the following for future work:

This is an active research area with novel solutions being explored, including multi-layer normalization strategies (Lin and Lu 2022), and the use of generative models (Pernice et al. 2023). Other potential solutions may include invariant risk minimization games (Ahuja et al. 2020) or similar formulations.

14. The Methods section is in many places too conceptual and philosophical. One wants concrete implementation details and specifics here.

We aimed to include rationale and explanation, which some reviewers appreciated, without making the main text overly detailed. We edited the Methods section but mainly kept this information in while adding implementation details and specifics where missing.

15. Methods 3.1: Is the compression really an important, central part of the method? It seems pretty convoluted, for an overall gain of a factor of 6, which is well within the order of magnitude of typical variations in dataset size, or the variability of available hardware resources in different labs. To me it seems more like a one-off hack.

Compression is not a central part of the method. The main concern with compression is that important biological information could be discarded. For this reason, we are transparent about reporting these choices. We added the following clarification to the methods section:

Note that compression is only used to optimize storage space, data transferring and computing time, especially given the large size of uncompressed datasets (which can be in the order of TBs).

16. “Stretch the histogram of intensities of each image by removing pixels that are too dark or too bright (below 0.05 and above 99.95 percentiles).” This is self-contradictory. A fixed number of pixels is removed (0.1%), whether these, and only these, are “too dark” or “too bright” is dataset-dependent, and a more data-adaptive threshold (if any) should be used.

Thanks for pointing this out. Following the spirit of not discarding important information before compression, we do not use adaptive thresholds to prevent distorting the original pixel distribution before quantization. Adaptive thresholds can result in major and unpredictable visual changes, especially in the presence of noise. We clarify this in the Methods section as follows:

Histogram clipping is not meant to enhance image contents, it is used as a preprocessing step before compressing pixel depth from 16 bits (64K pixel values) down to 8 bits (256 pixel values). Histogram clipping reduces the impact of noise in the tails of the original pixel distribution and to prevent allocation of compressed bits on unnecessarily high or low levels of brightness. The percentiles are relative to each individual image.

17. Methods 3.3: “Random crop and resize”—why is this appropriate? It seems to imply that variations in cell size are not biologically informative. While in fact the phenotypes of many cell cycle or metabolic genes comprise such variation and are biologically informative.

The reviewer is correct that cell size is an important phenotype in many biological processes and its variation is informative. We agree that random crop and resize may result in absolute cell size being down-weighted by the models. However, changes in cell size also influence other cellular morphology features, which may not be trivial to extract and we are interested in capturing. We clarify in the Methods section as follows:

Data augmentation is a strategy for regularization; it is used to prevent models from learning simple associations and to force the model to find alternative explanations. Crop and resize may change cell size, which is a biologically meaningful feature. However, it is not the only feature that we aim to capture in Cell Painting studies. Other simpler assays and algorithms can be used for that purpose. Instead, Cell Painting captures a wider range of structural and morphological cellular variations that are subtle and important for distinguishing the effects of thousands of perturbations. In practice, we observed that random cropping and resizing improves training performance and prevents overfitting. This is because treatments that impact cell size also tend to impact other features as a result, including colocalization of stains and textures. These features are then captured robustly when random variations in cell size are introduced as an augmentation during training.

18. “Random horizontal flips and 90-degree rotations.”. Why not general rotations (by any angle)? Just singling out these specific symmetry operators seems grossly incomplete.

The main reason for not using 360 degree rotations is that it introduces diagonal artifacts in images. In our work, single cells are segmented to identify their location, but we do not mask their context. Masking cells results in too much information being lost and performance degrading significantly (see Supplementary Figure 3). Generating a cropped cell in context without diagonal artifacts requires rotating the original, much larger image, which is computationally inefficient. In early development experiments, we found that neither of these solutions is satisfactory, and provides no benefits to the downstream tasks. Data augmentation can be as complex as it can get (even using generative models), but in our study we aimed to keep it simple.

19. “Random brightness and contrast adjustments, each channel is augmented and renormalized separately.”—again this seems to imply that brightness is not biologically informative, whereas experience from many cellular screens is that important phenotypes are affecting it.

The reviewer is right that brightness is an important biological readout. Unfortunately, absolute illumination values are also affected by technical artifacts, and the preprocessing steps do not correct all types of batch effects (plate-to-plate or well-to-well illumination variation). We therefore aim to learn intensity features that are robust to random brightness variations by allowing augmentations to vary the absolute illumination levels. We clarify this in the Methods section as follows:

Introducing random brightness and contrast augmentations prevents overfitting to non-biological illumination variations, resulting in more robust morphological features. The key observation is that pixel brightness is meaningful relative to other structural or morphological patterns in the same channel, but not in absolute values. By randomly changing the absolute brightness and contrast of each channel individually, we simulate

technical artifacts that could result in one of the channels having abnormal illumination variation (either brighter or darker than usual). Note that all the pixels in the same channel change by the same amount. This facilitates learning features that describe structural patterns in the image regardless of potentially unexpected illumination changes.

20. (Personally I also find such data augmentation ugly compared to using equivariance, but I understand “it’s what people do”.)

We understand the opinion of the reviewer. We added to the manuscript the possibility of investigating equivariance for cellular morphology in future work:

Instead of data augmentations, equivariance can be used as part of the model, which has been investigated in other domains, including pathology. We leave this possibility for future research.

21. Methods 3.5: “As ... images are five-channel and ImageNet pre-trained models expect three-channel (RGB) images, we follow the well established practice of computing a pseudo RGB image for each grayscale fluorescent channel by replicating it three times before passing it through the model (thus each cell requires five inference passes).” — This seems very ugly. Can you provide citations for the “well established practice”? Doesn’t this waste the opportunity of using correlations between the channels? Instead, e.g., why not use all possible 10 subsets of size 3 of the set of 5 channels (5 choose 3 is 10)? That way, the network would at least have the chance to directly access correlations between channels.

The reviewer is correct that computing features for each channel independently results in missing features from correlations across channels. However, this approach is more effective than forcing combinations of channels into a neural network trained for RGB images. Computing features from three channels at a time does not work because the joint pixel distribution of RGB images is drastically different from fluorescent channels. First, the way RGB images are normalized is based on the mean of pixel intensities in a natural image dataset, which is known to be biased towards uniform gray (gray world assumption in image enhancement literature). Instead, fluorescence images have a lower intensity mean, which is also different from channel to channel. When three fluorescent channels are normalized in this way, the resulting pixel intensities are out of distribution with respect to what the neural network expects. Second, forwarding three unevenly normalized channels through an RGB pre-trained neural network fails to capture the biological differences and correspondences between them. RGB pre-trained networks do not understand colors in the fluorescence microscopy world, resulting in features that only partially describe what the cells contain.

These mismatches between RGB images and microscopy images were observed early on when researchers attempted to use transfer learning for image-based profiling. Therefore, processing one channel at a time was proposed as a way to mitigate these issues and preserve the features of each channel at least separately. The approach has been used in various

image-based profiling studies that include the following: (Caicedo et al. 2022; Pawlowski et al. 2016; Michael Ando, McLean, and Berndl 2017; Ashdown et al. 2020; Qian et al. 2020; Cross-Zamirski et al. 2022). The approach is known to have two disadvantages: first, as the reviewer notes, the correlations between channels are missed. Second, the dimensionality of the resulting concatenated features grows linearly with the number of channels. Unfortunately, the suggestion by the reviewer of exploring more combinations would result in combinatorial dimensionality growth, in addition to not capturing more biologically meaningful features due to the differences between fluorescence and RGB images.

22. Methods 4.2: “The feature extraction steps are described in the CellProfiler pipelines published together with the corresponding original datasets. “—Sorry for the harshness but this sounds lazy. Please describe what was done here in a self-contained form, perhaps helped by a table. Where appropriate, provide specific citations.

We have rewritten the section, and added more details as recommended:

CellProfiler (Carpenter et al. 2006) allows the construction of customizable automated pipelines for analysis of biological images. It facilitates the analysis and extraction of meaningful information from high-throughput imaging experiments. The pipeline starts with image import, then it is followed by image pre-processing, such as illumination correction and noise removal. Then the objects are identified: nuclei are identified first and with this prior information the boundaries of the whole cell are inferred. Once cells are segmented, CellProfiler extracts feature vectors per cell, which are designed to be human readable and grouped by cell region (nucleus, cytoplasm or cell). Each of those feature groups has several common subgroups, such as shape features, intensity-based features, texture features and context features. In the table below there is a list of those feature subgroups.

Table - List of CellProfiler features used for profiling cellular morphology in our experiments.

Feature Group	Features and description.
AreaShape	Compactness, Eccentricity, Extent, FormFactor, MajorAxisLength, MaxFerretDiameter, MaximumRadius, MeanRadius, MedianRadius, MinFerretDiameter, MinorAxisLength, Orientation, Perimeter, Solidity, Zernike shape features.
Correlation, Correlation_Manders, Correlation_RWC	Those correlations are calculated for pairs of channels.
Granularity	Granularity for each channel and each instance of the granularity spectrum.
Intensity	IntegratedIntensityEdge, LowerQuartileIntensity, MADIntensity, MassDisplacement, MaxIntensityEdge, MeanIntensity,

	MedianIntensity, MinIntensity, StdIntensityEdge, StdIntensity, UpperQuartileIntensity. Each feature is calculated per channel.
Location	CenterMassIntensity, MaxIntensity. Each location is calculated per channel.
Neighbors	AngleBetweenNeighbors, FirstClosestDistance, NumberOfNeighbors, PercentTouching, SecondClosestObjectNumber. Features in this group are only calculated for “Cell” and “Nuclei” groups.
RadialDistribution	FracAtD, MeanFrac, RadialCV. Each feature was calculated per channel.
Texture	AngularSecondMoment, Contrast, Correlation, DifferenceEntropy, DifferenceVariance, Entropy, Gabor, InfoMeas1\2, InverseDifferenceMoment, SumAverage, SumEntropy, SumVariance, Variance. Each feature was calculated per channel.

Feature extraction for the evaluated datasets was performed with CellProfiler version 2. The table above lists the groups and names of features used in our experiments. Most of these features are extracted from each channel independently, except for the correlation features. The resulting size of single-cell feature vectors is approximately 1,700 (the exact value can vary among datasets). In our analysis, we used well-level aggregated profiles (see also “Feature aggregation and profiling”) to obtain baseline results. We reused publicly available features that were computed by the authors of the original study (BBBC037 (Rohban et al. 2019), BBBC022 (Gustafsdottir et al. 2013) and BBBC036 (Bray et al. 2017)). The links to the original data sources are listed in the Data Availability section below.

23. Methods 4.3: “Features of single-cells are first aggregated using the median operator at field-of-view (image) level. Next, fields-of-view features are aggregated using the mean to create a well-level profile. Finally, treatment-level profiles are obtained with the average across replicate wells.” — why this combination of medians and means? Why not always median or always mean? This is one of several places where the chosen approach seems to contain arbitrary choices (i.e. was not optimized). Maybe it does not matter, but the way it’s implemented seems a bit chaotic.

We clarify the rationale for this choice in the Methods section as follows:

This choice of alternating between median and mean follows the fact that at the field-of-view level there is more data (hundreds of cells), which is potentially noisy and prone to errors (from segmentation or feature extraction). Therefore the median is used as the robust estimator of the local morphological trend. At the replicate level, there are

fewer than ten data points to aggregate (9 fields of view, or 5 replicates), and then the average is computed to capture a smooth trend of morphological features among them. This pipeline was originally proposed by (Ljosa et al. 2013), and has been used in many other studies (Michael Ando, McLean, and Berndl 2017).

24. Methods 4.4: “Batch correction with the sphering transform”—is this more than just a clumsy way to implement a Mahalanobis distance (perhaps with some regularization)? In any case this seems like textbook stuff and should be appropriately cited.

This transformation was originally studied for image-based profiling in detail by (Michael Ando, McLean, and Berndl 2017), and has been subsequently used in many other studies (Way et al. 2022b; Caicedo et al. 2022; Lin and Lu 2022; Caicedo et al. 2018; Perakis et al. 2021; Janssens et al. 2021; Moshkov et al. 2023; Lippeveld et al. 2022; Tong et al. 2023; Haslum et al. 2022). We agree with the reviewer that the learned transformation can be interpreted as learning a Mahalanobis distance, with control samples as the reference population. We present a summary of our implementation of this technique and added citations to the original study and additional basic references to the sphering transform (Kessy, Lewin, and Strimmer 2018; Krizhevsky and Hinton 2009; Scott 1992).

References

- Ahuja, Kartik, Karthikeyan Shanmugam, Kush R. Varshney, and Amit Dhurandhar. 2020. "Invariant Risk Minimization Games." *arXiv [cs.LG]*. arXiv. <http://arxiv.org/abs/2002.04692>.
- Ashdown, George W., Michelle Dimon, Minjie Fan, Fernando Sánchez-Román Terán, Kathrin Witmer, David C. A. Gaboriau, Zan Armstrong, D. Michael Ando, and Jake Baum. 2020. "A Machine Learning Approach to Define Antimalarial Drug Action from Heterogeneous Cell-Based Screens." *Science Advances* 6 (39). <https://doi.org/10.1126/sciadv.aba9338>.
- Bao, Feng, Yue Deng, Sen Wan, Susan Q. Shen, Bo Wang, Qionghai Dai, Steven J. Altschuler, and Lani F. Wu. 2022. "Integrative Spatial Analysis of Cell Morphologies and Transcriptional States with MUSE." *Nature Biotechnology* 40 (8): 1200–1209.
- Bray, Mark-Anthony, Sigrun M. Gustafsdottir, Mohammad H. Rohban, Shantanu Singh, Vebjorn Ljosa, Katherine L. Sokolnicki, Joshua A. Bittker, et al. 2017. "A Dataset of Images and Morphological Profiles of 30 000 Small-Molecule Treatments Using the Cell Painting Assay." *GigaScience* 6 (12): 1–5.
- Caicedo, Juan C., John Arevalo, Federica Piccioni, Mark-Anthony Bray, Cathy L. Hartland, Xiaoyun Wu, Angela N. Brooks, et al. 2022. "Cell Painting Predicts Impact of Lung Cancer Variants." *Molecular Biology of the Cell* 33 (6): ar49.
- Caicedo, Juan C., Claire McQuin, Allen Goodman, Shantanu Singh, and Anne E. Carpenter. 2018. "Weakly Supervised Learning of Single-Cell Feature Embeddings." *Proceedings / CVPR, IEEE Computer Society Conference on Computer Vision and Pattern Recognition. IEEE Computer Society Conference on Computer Vision and Pattern Recognition 2018* (June): 9309–18.
- Carpenter, Anne E., Thouis R. Jones, Michael R. Lamprecht, Colin Clarke, In Han Kang, Ola Friman, David A. Guertin, et al. 2006. "CellProfiler: Image Analysis Software for Identifying and Quantifying Cell Phenotypes." *Genome Biology* 7 (10): R100.
- Cross-Zamirski, Jan Oscar, Guy Williams, Elizabeth Mouchet, Carola-Bibiane Schönlieb, Riku Turkki, and Yin Hai Wang. 2022. "Self-Supervised Learning of Phenotypic Representations from Cell Images with Weak Labels." *arXiv [cs.CV]*. arXiv. <http://arxiv.org/abs/2209.07819>.
- Cuccarese, Michael F., Berton A. Earnshaw, Katie Heiser, Ben Fogelson, Chadwick T. Davis, Peter F. McLean, Hannah B. Gordon, et al. 2020. "Functional Immune Mapping with Deep-Learning Enabled Phenomics Applied to Immunomodulatory and COVID-19 Drug Discovery." <https://doi.org/10.1101/2020.08.02.233064>.
- Ganin, Yaroslav, and Victor Lempitsky. 2015. "Unsupervised Domain Adaptation by Backpropagation." In *Proceedings of the 32nd International Conference on Machine Learning*, edited by Francis Bach and David Blei, 37:1180–89. Proceedings of Machine Learning Research. Lille, France: PMLR.
- Gustafsdottir, Sigrun M., Vebjorn Ljosa, Katherine L. Sokolnicki, J. Anthony Wilson, Deepika Walpita, Melissa M. Kemp, Kathleen Petri Seiler, et al. 2013. "Multiplex Cytological Profiling Assay to Measure Diverse Cellular States." *PloS One* 8 (12): e80999.
- Haslum, Johan Fredin, Christos Matsoukas, Karl-Johan Leuchowius, Erik Müllers, and Kevin Smith. 2022. "Metadata-Guided Consistency Learning for High Content Images." *arXiv [cs.CV]*. arXiv. <http://arxiv.org/abs/2212.11595>.
- Hofmarcher, Markus, Elisabeth Rumetshofer, Djork-Arné Clevert, Sepp Hochreiter, and Günter Klambauer. 2019. "Accurate Prediction of Biological Assays with High-Throughput Microscopy Images and Convolutional Networks." *Journal of Chemical*. <https://pubs.acs.org/doi/abs/10.1021/acs.jcim.8b00670>.

- Ilse, Maximilian, Jakub Tomczak, and Max Welling. 2018. "Attention-Based Deep Multiple Instance Learning." In *Proceedings of the 35th International Conference on Machine Learning*, edited by Jennifer Dy and Andreas Krause, 80:2127–36. Proceedings of Machine Learning Research. PMLR.
- Janssens, Rens, Xian Zhang, Audrey Kauffmann, Antoine de Weck, and Eric Y. Durand. 2021. "Fully Unsupervised Deep Mode of Action Learning for Phenotyping High-Content Cellular Images." *Bioinformatics* 37 (23): 4548–55.
- Kensert, Alexander, Philip J. Harrison, and Ola Spjuth. 2019. "Transfer Learning with Deep Convolutional Neural Networks for Classifying Cellular Morphological Changes." *SLAS Discovery : Advancing Life Sciences R & D* 24 (4): 466–75.
- Kessy, Agnan, Alex Lewin, and Korbinian Strimmer. 2018. "Optimal Whitening and Decorrelation." *The American Statistician* 72 (4): 309–14.
- Kim, Vladislav, Nikolaos Adaloglou, Marc Osterland, Flavio M. Morelli, and Paula A. Marin Zapata. 2023. "Self-Supervision Advances Morphological Profiling by Unlocking Powerful Image Representations." *bioRxiv*. <https://doi.org/10.1101/2023.04.28.538691>.
- Korsunsky, Ilya, Nghia Millard, Jean Fan, Kamil Slowikowski, Fan Zhang, Kevin Wei, Yuriy Baglaenko, Michael Brenner, Po-Ru Loh, and Soumya Raychaudhuri. 2019. "Fast, Sensitive and Accurate Integration of Single-Cell Data with Harmony." *Nature Methods* 16 (12): 1289–96.
- Krizhevsky, A., and G. Hinton. 2009. "Learning Multiple Layers of Features from Tiny Images." <http://citeseerx.ist.psu.edu/viewdoc/download?doi=10.1.1.222.9220&rep=rep1&type=pdf>.
- Lin, Alexander, and Alex Lu. 2022. "Incorporating Knowledge of Plates in Batch Normalization Improves Generalization of Deep Learning for Microscopy Images." In *Proceedings of the 17th Machine Learning in Computational Biology Meeting*, edited by David A. Knowles, Sara Mostafavi, and Su-In Lee, 200:74–93. Proceedings of Machine Learning Research. PMLR.
- Lippeveld, Maxim, Daniel Peralta, Andrew Filby, and Yvan Saeys. 2022. "A Scalable, Reproducible and Open-Source Pipeline for Morphologically Profiling Image Cytometry Data." *bioRxiv*. <https://doi.org/10.1101/2022.10.24.512549>.
- Li, Xingyu, Min Cen, Jinfeng Xu, Hong Zhang, and Xu Steven Xu. 2022. "Improving Feature Extraction from Histopathological Images through a Fine-Tuning ImageNet Model." *Journal of Pathology Informatics* 13 (June): 100115.
- Ljosa, Vebjorn, Peter D. Caie, Rob Ter Horst, Katherine L. Sokolnicki, Emma L. Jenkins, Sandeep Daya, Mark E. Roberts, et al. 2013. "Comparison of Methods for Image-Based Profiling of Cellular Morphological Responses to Small-Molecule Treatment." *Journal of Biomolecular Screening* 18 (10): 1321–29.
- Lu, Alex X., Oren Z. Kraus, Sam Cooper, and Alan M. Moses. 2019. "Learning Unsupervised Feature Representations for Single Cell Microscopy Images with Paired Cell inpainting." *PLoS Computational Biology* 15 (9): e1007348.
- Mao, Chengzhi, Augustine Cha, Amogh Gupta, Hao Wang, Junfeng Yang, and Carl Vondrick. 2021. "Generative Interventions for Causal Learning." In *2021 IEEE/CVF Conference on Computer Vision and Pattern Recognition (CVPR)*. IEEE. <https://doi.org/10.1109/cvpr46437.2021.00394>.
- Michael Ando, D., Cory Y. McLean, and Marc Berndt. 2017. "Improving Phenotypic Measurements in High-Content Imaging Screens." *bioRxiv*. <https://doi.org/10.1101/161422>.
- Moshkov, Nikita, Tim Becker, Kevin Yang, Peter Horvath, Vlado Dancik, Bridget K. Wagner, Paul A. Clemons, Shantanu Singh, Anne E. Carpenter, and Juan C. Caicedo. 2023. "Predicting Compound Activity from Phenotypic Profiles and Chemical Structures." *Nature Communications* 14 (1): 1967.
- Pawlowski, Nick, Juan C. Caicedo, Shantanu Singh, Anne E. Carpenter, and Amos Storkey. 2016. "Automating Morphological Profiling with Generic Deep Convolutional Networks."

- bioRxiv*. <https://doi.org/10.1101/085118>.
- Perakis, Alexis, Ali Gorji, Samridhi Jain, Krishna Chaitanya, Simone Rizza, and Ender Konukoglu. 2021. "Contrastive Learning of Single-Cell Phenotypic Representations for Treatment Classification." In *Machine Learning in Medical Imaging*, 565–75. Springer International Publishing.
- Pernice, Wolfgang M., Michael Doron, Alex Quach, Aditya Pratapa, Sultan Kenjeyev, Nicholas De Veaux, Michio Hirano, and Juan C. Caicedo. 2023. "Out of Distribution Generalization via Interventional Style Transfer in Single-Cell Microscopy." In *Proceedings of the IEEE/CVF Conference on Computer Vision and Pattern Recognition*, 4325–34.
- Qian, Wesley Wei, Cassandra Xia, Subhashini Venugopalan, Arunachalam Narayanaswamy, Michelle Dimon, George W. Ashdown, Jake Baum, Jian Peng, and D. Michael Ando. 2020. "Batch Equalization with a Generative Adversarial Network." *Bioinformatics* 36 (Suppl_2): i875–83.
- Rohban, Mohammad H., Hamdah S. Abbasi, Shantanu Singh, and Anne E. Carpenter. 2019. "Capturing Single-Cell Heterogeneity via Data Fusion Improves Image-Based Profiling." *Nature Communications* 10 (1): 2082.
- Schiff, Lauren, Bianca Migliori, Ye Chen, Deidre Carter, Caitlyn Bonilla, Jenna Hall, Minjie Fan, et al. 2022. "Integrating Deep Learning and Unbiased Automated High-Content Screening to Identify Complex Disease Signatures in Human Fibroblasts." *Nature Communications* 13 (1): 1590.
- Schölkopf, Bernhard, Francesco Locatello, Stefan Bauer, Nan Rosemary Ke, Nal Kalchbrenner, Anirudh Goyal, and Yoshua Bengio. 2021. "Toward Causal Representation Learning." *Proceedings of the IEEE* 109 (5): 612–34.
- Scott, David W. 1992. *Multivariate Density Estimation: Theory, Practice, and Visualization*. John Wiley & Sons.
- Subramanian, Aravind, Rajiv Narayan, Steven M. Corsello, David D. Peck, Ted E. Natoli, Xiaodong Lu, Joshua Gould, et al. 2017a. "A Next Generation Connectivity Map: L1000 Platform and the First 1,000,000 Profiles." *Cell* 171 (6): 1437–52.e17.
- . 2017b. "A Next Generation Connectivity Map: L1000 Platform and the First 1,000,000 Profiles." *Cell* 171 (6): 1437–52.e17.
- Tong, Lei, Adam Corrigan, Navin Rathna Kumar, Kerry Hallbrook, Jonathan Orme, Yinhai Wang, and Huiyu Zhou. 2023. "CLANet: A Comprehensive Framework for Cross-Batch Cell Line Identification Using Brightfield Images." *arXiv [cs.CV]*. arXiv. <http://arxiv.org/abs/2306.16538>.
- Way, Gregory P., Ted Natoli, Adeniyi Adeboye, Lev Litichevskiy, Andrew Yang, Xiaodong Lu, Juan C. Caicedo, et al. 2022a. "Morphology and Gene Expression Profiling Provide Complementary Information for Mapping Cell State." *Cell Systems* 13 (11): 911–23.e9.
- . 2022b. "Morphology and Gene Expression Profiling Provide Complementary Information for Mapping Cell State." *Cell Systems* 13 (11): 911–23.e9.
- Wong, Daniel R., David J. Logan, Santosh Hariharan, Robert Stanton, Djork-Arné Clevert, and Andrew Kiruluta. 2023. "Deep Representation Learning Determines Drug Mechanism of Action from Cell Painting Images." *Digital Discovery*, August. <https://doi.org/10.1039/D3DD00060E>.
- Wong, Daniel R., David J. Logan, Santosh Hariharan, Robert Stanton, and Andrew Kiruluta. 2022. "Deep Representation Learning Determines Drug Mechanism of Action from Cell Painting Images." *bioRxiv*. <https://doi.org/10.1101/2022.11.15.516561>.
- Yang, Samuel J., Scott L. Lipnick, Nina R. Makhortova, Subhashini Venugopalan, Minjie Fan, Zan Armstrong, Thorsten M. Schlaeger, et al. 2019. "Applying Deep Neural Network Analysis to High-Content Image-Based Assays." *SLAS Discovery : Advancing Life Sciences R & D* 24 (8): 829–41.
- Zhang, Chang-Bin, Peng-Tao Jiang, Qibin Hou, Yunchao Wei, Qi Han, Zhen Li, and Ming-Ming

Cheng. 2021. "Delving Deep Into Label Smoothing." *IEEE Transactions on Image Processing: A Publication of the IEEE Signal Processing Society* 30 (June): 5984–96.

REVIEWERS' COMMENTS

Reviewer #1 (Remarks to the Author):

Dear authors,

Upon my review, I am pleased to report that you have adequately addressed the comments and concerns I raised in my initial review.

Thank you for your great work.

Reviewer #4 (Remarks to the Author):

The clarity and precision of the manuscript has much improved, and the authors addressed many of the suggestions I had made.

I do not think another round of commenting and revising would add much value to the ms, and would be happy to see this come out in the journal.